# Contextual Bandits and Imitation Learning with Preference-Based Active Queries

**Ayush Sekhari**[*]
MIT
sekhari@mit.edu

**Karthik Sridharan**
Cornell University
ks999@cornell.edu

**Wen Sun**
Cornell University
ws455@cornell.edu

**Runzhe Wu**
Cornell University
rw646@cornell.edu

## Abstract

We consider the problem of contextual bandits and imitation learning, where the learner lacks direct knowledge of the executed action's reward. Instead, the learner can actively request the expert at each round to compare two actions and receive noisy preference feedback. The learner's objective is two-fold: to minimize regret associated with the executed actions, while simultaneously, minimizing the number of comparison queries made to the expert. In this paper, we assume that the learner has access to a function class that can represent the expert's preference model under appropriate link functions and present an algorithm that leverages an online regression oracle with respect to this function class. For the contextual bandit setting, our algorithm achieves a regret bound that combines the best of both worlds, scaling as $O(\min\{\sqrt{T}, d/\Delta\})$, where $T$ represents the number of interactions, $d$ represents the eluder dimension of the function class, and $\Delta$ represents the minimum preference of the optimal action over any suboptimal action under all contexts. Our algorithm does not require the knowledge of $\Delta$, and the obtained regret bound is comparable to what can be achieved in the standard contextual bandits setting where the learner observes reward signals at each round. Additionally, our algorithm makes only $O(\min\{T, d^2/\Delta^2\})$ queries to the expert. We then extend our algorithm to the imitation learning setting, where the agent engages with an unknown environment in episodes of length $H$, and provide similar guarantees regarding regret and query complexity. Interestingly, with preference-based feedback, our imitation learning algorithm can learn a policy outperforming a sub-optimal expert, matching the result from interactive imitation learning algorithms [Ross and Bagnell, 2014] that require access to the expert's actions and also reward signals.

## 1 Introduction

Human feedback for training machine learning models has been widely used in scenarios including robotics [Ross et al., 2011, 2013, Jain et al., 2015, Laskey et al., 2016, Christiano et al., 2017] and natural language processing [Stiennon et al., 2020, Ouyang et al., 2022]. By integrating human feedback into the training process, these prior works provide techniques to align machine-learning models with human intention and enable high-quality human-machine interaction (e.g., ChatGPT).

---

[*]Authors are listed in alphabetical order of their last names.

37th Conference on Neural Information Processing Systems (NeurIPS 2023).

Existing methods generally leverage two types of human feedback. The first is the action from human experts, which is the dominant feedback mode in the literature of imitation learning and learning from demonstrations [Abbeel and Ng, 2004, Ziebart et al., 2008, Daumé et al., 2009, Ross et al., 2011, Ross and Bagnell, 2014, Sun et al., 2017, Osa et al., 2018, Li et al., 2023]. The second type of feedback is preference-based feedback, which involves comparing pairs of actions. In this approach, the expert provides feedback by indicating their preference between two options selected by the learner. While both types of feedback have their applications, our focus in this work is on preference-based feedback, which is particularly suitable for scenarios where it is challenging for human experts to recommend the exact optimal action while making pairwise comparisons is much easier.

Learning via preference-based feedback has been extensively studied, particularly in the field of *dueling bandits* [Yue and Joachims, 2011, Yue et al., 2012, Zoghi et al., 2014, Ailon et al., 2014, Komiyama et al., 2015, Wu and Liu, 2016, Saha and Gaillard, 2021, Bengs et al., 2021, Saha and Gaillard, 2022] and *contextual dueling bandits* [Dudík et al., 2015, Saha, 2021, Saha and Krishnamurthy, 2022, Bengs et al., 2022, Wu et al., 2023]. Different from the standard bandit setting, the learner proposes two actions in dueling bandits and only gets noisy preference feedback from the human expert. Follow-up works extend the preference-based learning model from the one-step bandit setting to the multi-step decision-making (e.g., imitation learning and reinforcement learning) setting [Chu and Ghahramani, 2005, Sadigh et al., 2017, Christiano et al., 2017, Lee et al., 2021b, Chen et al., 2022, Saha et al., 2023]. These studies mainly focus on learning a high-quality policy from human feedback, without concerning the question of active query in order to minimize the query complexity.

However, query complexity is an important metric to optimize when learning from human feedback, as human feedback is expensive to collect [Lightman et al., 2023]. For instance, InstructGPT [Ouyang et al., 2022] is trained only on around 30K pieces of human feedback, which is significantly fewer than the internet-scale dataset used for pre-training the base model GPT3, indicating the challenge of scaling up the size of human feedback datasets. In other areas, such as robotics, learning from human feedback is also not easy, and prior studies (e.g., Cohn et al. [2011], Zhang et al. [2022], Myers et al. [2023]) have explored this issue from various perspectives. Ross et al. [2013], Laskey et al. [2016] pointed out that querying human feedback in the learning loop is challenging, and extensively querying for feedback puts too much burden on the human experts.

In this work, we design *principled algorithms that learn from preference-based feedback while at the same time minimizing query complexity* under the settings of contextual bandits [Auer et al., 2002] and imitation learning [Ross et al., 2011]. Our main contributions can be summarized as follows.

- In the contextual dueling bandits setting, the stochastic preference feedback is generated based on some preference matrix [Saha and Krishnamurthy, 2022]. We propose an algorithm (named AURORA – in short of *Active preference qUeRy fOR contextual bAndits*) that achieves a best-of-both-worlds regret bound (i.e., achieving the minimum of the worst-case regret and an instance dependent regret), while at the same providing an instance-dependent query complexity bound. For benign instances with small eluder dimension and large gap, our regret and query complexity bounds both scale with $\ln(T)$ where $T$ is the total number of interactions in contextual bandits.

- In imitation learning, the stochastic preference feedback is generated based on the reward-to-go of the expert's policy (e.g., the expert prefers actions that lead to higher reward-to-go). We propose an algorithm named AURORAE, in short of *Active preference qUeRy fOR imitAtion lEarning*, which instantiates $H$ instances of AURORA, one per each time step for the finite horizon Markov Decision Process (MDP), where $H$ is the horizon. By leveraging preference-based feedback, we show that, interestingly, our algorithm can learn to outperform the expert when the expert is suboptimal. Such a result is beyond the scope of the classic imitation learning algorithm DAGGER and previously can only be achieved by algorithms like AGGREVATE(D) [Ross and Bagnell, 2014, Sun et al., 2017, Cheng and Boots, 2018] and LOLS [Chang et al., 2015] which require direct access to expert's actions and reward signal – a much stronger feedback mode than ours.

To the best of our knowledge, for both contextual bandit and imitation learning with preference-based feedback, our algorithms are the first to achieve best-of-both-worlds regret bounds via active querying.

## 1.1 Related works

**Selective Sampling.** Numerous studies have been conducted on selective sampling across various settings [Cesa-Bianchi et al., 2005, Dekel et al., 2012, Agarwal, 2013, Hanneke and Yang, 2015,

2021, Zhu and Nowak, 2022], with the work of Sekhari et al. [2023] being closest to ours. Sekhari et al. [2023] presented a suite of provably efficient algorithms that are applicable to settings including contextual bandits and imitation learning. The primary distinction between our setting and the prior works lies in the feedback modality–we assume preference-based feedback, whereas they assume direct label feedback or reward signals.

**Contextual bandits with preference feedback.** Dudík et al. [2015] is the first to consider contextual dueling bandits, and one of their algorithms achieves the optimal regret rate. Saha and Krishnamurthy [2022] studied contextual dueling bandits using a value function class and proposed an algorithm based on a reduction to online regression, which also achieves an optimal worst-case regret bound. In this paper, we mainly follow the setting of the latter and make notable improvements in two aspects: (1) in addition to the $O(\sqrt{AT})$ optimal regret rate where $A$ is the number of actions and $T$ is the number of interaction rounds, we established an instance-dependent regret upper bound that can be significantly smaller when the bandit exhibits a favorable structure; (2) our algorithm has an instance-dependent upper bound on the number of queries, and thus when the underlying instance is well behaved (has small eluder dimension and large gap), we will make significantly fewer queries.

Another related work is Saha and Gaillard [2022] which achieves the best-of-both-worlds regret for non-contextual dueling bandits. Our setting is more general due to the context and general function approximation, which enables us to leverage function classes beyond linear and tabular cases.

**RL with preference feedback.** RL with preference feedback has been widely employed in recent advancements in AI [Ouyang et al., 2022, OpenAI, 2023]. According to Wirth et al. [2017], there are generally three types of preference feedback: action preferences [Fürnkranz et al., 2012], state preferences [Wirth and Fürnkranz, 2014], and trajectory preferences [Busa-Fekete et al., 2014, Novoseller et al., 2020, Xu et al., 2020, Lee et al., 2021a, Chen et al., 2022, Saha et al., 2023, Pacchiano et al., 2021, Biyik and Sadigh, 2018, Taranovic et al., 2022, Sadigh et al., 2017]. We focus on the action preference with the goal of achieving tight regret bounds and query complexities.

The concurrent work from Zhan et al. [2023] investigates the experimental design in both the trajectories-based and action-based preference settings, for which they decouple the process of collecting trajectories from querying for human feedback. Their action-based setting is the same as ours, but they mainly focus on linear parameterization, while our approach is a reduction to online regression and can leverage general function approximation beyond linear function classes.

**Imitation learning.** In imitation learning, two common feedback modalities are typically considered: expert demonstration and preference. The former involves directly acquiring expert actions [Ross et al., 2011, Ross and Bagnell, 2014, Sun et al., 2017, Chang et al., 2015, Sekhari et al., 2023], while the latter focuses on obtaining preferences between selected options [Chu and Ghahramani, 2005, Lee et al., 2021b, Zhu et al., 2023]. Brown et al. [2019, 2020] leveraged both demonstrations and preference-based information and empirically showed that their algorithm can learn to outperform experts. Our imitation learning setting belongs to the second category, and we established bounds on the regret and the query complexity for our algorithm. We show that our algorithm can learn a policy that can provably outperform the expert (when it is suboptimal for the underlying environment).

## 2 Preliminaries

### 2.1 Contextual Bandits with Preference-Based Feedback

In this section, we introduce the contextual dueling bandits setting. For notation, we denote $[N]$ as the integer set $\{1, \ldots, N\}$ and denote $\Delta(\mathcal{S})$ as the set of distributions over a set $\mathcal{S}$.

We assume a context set $\mathcal{X}$ and an action space $\mathcal{A} = [A]$. At each round $t \in [T]$, a context $x_t$ is drawn *adversarially*, and the learner's task is to pick a pair of actions $(a_t, b_t) \in \mathcal{A} \times \mathcal{A}$ and then decide whether to make a query to the expert. If making a query, a noisy feedback $y_t \in \{-1, 1\}$ is revealed to the learner regarding whether $a_t$ or $b_t$ is better. We assume that the expert relies on a preference function $f^\star : \mathcal{X} \times \mathcal{A} \times \mathcal{A} \to [-1, 1]$ to samples its preference feedback $y_t$:

$$\Pr(a_t \text{ is preferred to } b_t \mid x_t) \coloneqq \Pr(y_t = 1 \mid x_t, a_t, b_t) = \phi\big(f^\star(x_t, a_t, b_t)\big)$$

where $\phi(d) : [-1, 1] \to [0, 1]$ is the link function, which satisfies $\phi(d) + \phi(-d) = 1$ for any $d$. If the learner does not make a query, it will not receive any feedback for the selected actions $a_t$ and $b_t$. Let $Z_t \in \{0, 1\}$ indicate whether the learner makes a query at round $t$.

We assume that the learner has access to a function class $\mathcal{F} \subseteq \mathcal{X} \times \mathcal{A} \times \mathcal{A} \to [-1, 1]$ that realizes $f^\star$. Furthermore, we assume that $f^\star$, as well as the functions in $\mathcal{F}$, is transitive and anti-symmetric.

**Assumption 1.** *We assume $f^\star \in \mathcal{F}$ and any functions $f \in \mathcal{F}$ satisfies the following two properties: (1) transitivity: for any $x \in \mathcal{X}$ and $a, b, c \in \mathcal{A}$, if $f(x, a, b) > 0$ and $f(x, b, c) > 0$, then we must have $f(x, a, c) > 0$; (2) anti-symmetry: $f(x, a, b) = -f(x, b, a)$ for any $x \in \mathcal{X}$ and any $a, b \in \mathcal{A}$.*

We provide an example below for which Assumption 1 is satisfied.

**Example 1.** *Assume there exists a function $r^\star : \mathcal{X} \times \mathcal{A} \to [0, 1]$ such that $f^\star(x, a, b) = r^\star(x, a) - r^\star(x, b)$ for any $x \in \mathcal{X}$ and $a, b \in \mathcal{A}$. Typically, such a function $r^\star$ represents the "reward function" of the contextual bandit. In such a scenario, we can first parameterize a reward class $\mathcal{R} \subseteq \mathcal{X} \times \mathcal{A} \to [0, 1]$ and define $\mathcal{F} = \{f : f(x, a, b) = r(x, a) - r(x, b), r \in \mathcal{R}\}$. Moreover, it is common to have $\phi(d) := 1/(1 + \exp(-d))$ in this setting, which recovers the Bradley-Terry-Luce (BTL) model [Bradley and Terry, 1952] — a commonly used model in practice for learning reward models [Christiano et al., 2017].*

Assumption 1 ensures the existence of an optimal arm, as stated below.

**Lemma 1.** *Under Assumption 1, for any function $f \in \mathcal{F}$ and any context $x \in \mathcal{X}$, there exists an arm $a \in \mathcal{A}$ such that $f(x, a, b) \geq 0$ for any arm $b \in \mathcal{A}$. We denote this best arm by $\pi_f(x) := a$.*[2]

The learner's goal is to minimize the regret while also minimizing the number of queries, defined as:

$$\text{Regret}_T^{\text{CB}} := \sum_{t=1}^{T} \left( f^\star(x_t, \pi_{f^\star}(x_t), a_t) + f^\star(x_t, \pi_{f^\star}(x_t), b_t) \right), \quad \text{Queries}_T^{\text{CB}} := \sum_{t=1}^{T} Z_t.$$

It is worth noting that when $f^\star$ is the difference in rewards (as in Example 1), the regret defined above reduces to the standard regret of a contextual bandit. We also remark that our feedback model generalizes that of Saha and Krishnamurthy [2022] in that we assume an additional link function $\phi$, while they assume the feedback is sampled from $\Pr(y = 1 \mid x, a, b) = (P_t[a_t, b_t] + 1)/2$ where $P_t$ is a preference matrix. Their loss function is captured in our setting (Example 2). However, Saha and Krishnamurthy [2022] do not assume transitivity.

## 2.2 Imitation Learning with Preference-Based Feedback

In our imitation learning setup, we consider that the learner operates in a finite-horizon Markov decision process (MDP), which is a tuple $M(\mathcal{X}, \mathcal{A}, r, P, H)$ where $\mathcal{X}$ is the state space, $\mathcal{A}$ is the action space, $P$ is the transition kernel, $r : \mathcal{X} \times \mathcal{A} \to [0, 1]$ is the reward function, and $H$ is the length of each episode. The interaction between the learner and the environment proceeds as follows: at each episode $t \in [T]$, the learner receives an initial state $x_{t,0}$ which could be chosen adversarially. Then, the learner interacts with the environment for $H$ steps. At each step $h$, the learner first decides whether to make a query. If making a query, the learner needs to select a pair of actions $(a_{t,h}, b_{t,h}) \in \mathcal{A} \times \mathcal{A}$, upon which a feedback $y_{t,h} \in \{-1, 1\}$ is revealed to the learner regarding which action is preferred from the expert's perspective. Here the feedback is sampled according to

$$\Pr(a_{t,h} \text{ is preferred to } b_{t,h} \mid x_{t,h}, h) := \Pr(y_{t,h} = 1 \mid x_{t,h}, a_{t,h}, b_{t,h}, h) = \phi\left(f_h^\star(x, a_{t,h}, b_{t,h})\right).$$

Irrespective of whether the learner made a query, it then picks a single action from $a_{t,h}, b_{t,h}$ and transit to the next step (our algorithm will just pick an action uniformly at random from $a_{t,h}, b_{t,h}$). After $H$ steps, the next episode starts. Let $Z_{t,h} \in \{0, 1\}$ indicate whether the learner decided to query at step $h$ in episode $t$. We assume that the function class $\mathcal{F}$ is a product of $H$ classes, i.e., $\mathcal{F} = \mathcal{F}_0 \times \cdots \mathcal{F}_{H-1}$ where, for each $h$, we use $\mathcal{F}_h = \{f : \mathcal{X} \times \mathcal{A} \times \mathcal{A} \to [-1, 1]\}$ to model $f_h^\star$ and assume that $\mathcal{F}_h$ satisfies Assumption 1.

A policy is a mapping $\pi : \mathcal{X} \to \Delta(\mathcal{A})$. For a policy $\pi$, the state value function for a state $x$ at step $h$ is defined as $V_h^\pi(x) := \mathbb{E}[\sum_{i=h}^{H-1} r_i \mid x_h = x]$ and the state-action value function for a state-action pair $(x, a)$ is $Q_h^\pi(x, a) := \mathbb{E}[\sum_{i=h}^{H-1} r_i \mid x_h = x, a_h = a]$, where the expectations are taken w.r.t. the trajectories sampled by $\pi$ in the underlying MDP.

In the imitation learning setting, we assume that the expert (who gives the preference-based feedback) is equipped with a markovian policy $\pi_e$, and that the preference of the expert is dependent on the

---

[2]When the best arms is not unique, the ties are broken arbitrarily but consistently.

reward-to-go under $\pi_e$ (i.e. on a state $x$, actions with higher values of $Q^{\pi_e}(s,a)$ will be preferred by the expert). Formalizing this intuition, we assume that $f_h^\star$ is defined such that as $f_h^\star(x,a,b) := Q_h^{\pi_e}(x,a) - Q_h^{\pi_e}(x,b)$. The goal of the learner is still to minimize the regret and number of queries:

$$\text{Regret}_T^{\text{IL}} := \sum_{t=1}^{T} \left( V_0^{\pi_e}(x_{t,0}) - V_0^{\pi_t}(x_{t,0}) \right), \quad \text{Queries}_T^{\text{IL}} := \sum_{t=1}^{T} \sum_{h=0}^{H-1} Z_{t,h}.$$

Here $\pi_t$ is the strategy the learner uses to select actions at episode $t$.

## 2.3 Link Function and Online Regression Oracle

Following the standard practice in the literature [Agarwal, 2013], we assume $\phi$ is the derivative of some $\alpha$-strongly convex function (Definition 3) $\Phi : [-1,1] \to \mathbb{R}$ and define the associated loss function as $\ell_\phi(d,y) = \Phi(d) - d(y+1)/2$. Additionally, in line with prior works [Foster et al., 2021, Foster and Rakhlin, 2020, Simchi-Levi and Xu, 2022, Foster et al., 2018a, Sekhari et al., 2023], our algorithm utilizes an online regression oracle, which is assumed to have a sublinear regret guarantee w.r.t. $\mathcal{F}$ on arbitrary data sequences.

**Assumption 2.** *We assume the learner has access to an online regression oracle pertaining to the loss $\ell_\phi$ such that for any sequence $\{(x_1,a_1,b_1,y_1),\ldots,(x_T,a_T,b_T,y_T)\}$ where the label $y_t$ is generated by $y_t \sim \phi(f^\star(x_t,a_t,b_t))$, we have*

$$\sum_{t=1}^{T} \ell_\phi\big(f_t(x_t,a_t,b_t),y_t\big) - \inf_{f \in \mathcal{F}} \ell_\phi\big(f(x_t,a_t,b_t),y_t\big) \le \Upsilon(\mathcal{F},T)$$

*for some $\Upsilon(\mathcal{F},T)$ that grows sublinearly with respect to $T$.[3] For notational simplicity, whenever clear from the context, we define $\Upsilon := \Upsilon(\mathcal{F},T)$.*

Here $\Upsilon$ represents the regret upper bound and is typically of logarithmic order in $T$ or the cardinality of the function class $\mathcal{F}$ in many cases (here we drop the dependence on $T$ in notation for simplicity). We provide a few examples below.

**Example 2** (Squared loss). *If we consider $\Phi(d) = d^2/4 + d/2 + 1/4$, which is $1/4$-strongly convex, then we obtain $\phi(d) = (d+1)/2$ and $\ell_\phi(d,y) = (d-y)^2/4$, thereby recovering the squared loss, which has been widely studied in prior works. For example, Rakhlin and Sridharan [2014] characterized the minimax rates for online square loss regression in terms of the offset sequential Rademacher complexity, resulting in favorable bounds for the regret. Specifically, we have $\Upsilon = O(\log|\mathcal{F}|)$ assuming the function class $\mathcal{F}$ is finite, and $\Upsilon = O(d\log(T))$ assuming $\mathcal{F}$ is a $d$-dimensional linear class. We also kindly refer the readers to Krishnamurthy et al. [2017], Foster et al. [2018a] for efficient implementations.*

**Example 3** (Logistic loss). *When $\Phi(d) = \log(1+\exp(d))$ which is strongly convex at $[-1,1]$, we have $\phi(d) = 1/(1+\exp(-d))$ and $\ell_\phi(d,y) = \log(1+\exp(-yd))$. Thus, we recover the logistic regression loss, which allows us to use online logistic regression and achieve $\Upsilon = O(\log|\mathcal{F}|)$ assuming finite $\mathcal{F}$. There have been numerous endeavors in minimizing the log loss, such as Foster et al. [2018b] and Cesa-Bianchi and Lugosi [2006, Chapter 9].*

# 3 Contextual Bandits with Preference-Based Active Queries

We first present the algorithm, named AURORA, for contextual dueling bandits, as shown in Algorithm 1. At each round $t \in [T]$, the learner first constructs a version space $\mathcal{F}_t$ containing all functions close to past predictors on the observed data. Here, the threshold $\beta$ set to $4\Upsilon/\alpha + (16 + 24\alpha)\log\big(4\delta^{-1}\log(T)\big)/\alpha^2$ ensures that $f^\star \in \mathcal{F}_t$ for any $t \in [T]$ with probability at least $1 - \delta$ (Lemma 9). Thus, $\mathcal{A}_t$ is non-empty for all $t \in [T]$ and correspondingly Line 17 is well defined. The learner then forms a candidate arm set $\mathcal{A}_t$ consisting of greedy arms induced by all functions in the version space. When $|\mathcal{A}_t| = 1$, the only arm in the set is the optimal arm since $f^\star \in \mathcal{F}_t$, and thus no query is needed ($Z_t = 0$). However, when $|\mathcal{A}_t| > 1$, any arm in $\mathcal{A}_t$ could potentially be the optimal arm, and thus the learner needs to make a comparison query to obtain more information.

---

[3]The online regression oracle updates as follows: in each iteration, after seeing $x_t, a_t, b_t$, it proposes a decision $f_t$, then $y_t$ is revealed and the online regression oracle incurs loss $\ell_\phi(f_t(x_t,a_t,b_t),y_t)$.

---

**Algorithm 1** Active preference qUeRy fOR contextual bAndits (AURORA)

---

**Require:** Function class $\mathcal{F}$, confidence parameter $\beta = \frac{4\Upsilon}{\alpha} + \frac{16+24\alpha}{\alpha^2} \log\left(4\delta^{-1}\log(T)\right)$.

1: Online regression oracle produces $f_1$.
2: **for** $t = 1, 2, \ldots, T$ **do**
3:     Learner receives context $x_t$, computes the version space

$$\mathcal{F}_t \leftarrow \left\{ f \in \mathcal{F} : \sum_{s=1}^{t-1} Z_s \Big( f(x_s, a_s, b_s) - f_s(x_s, a_s, b_s) \Big)^2 \leq \beta \right\}.$$

      and the candidate arm set $\mathcal{A}_t \leftarrow \{\pi_f(x_t) : \forall f \in \mathcal{F}_t\}$.
4:     Learner decides whether to query $Z_t \leftarrow \mathbb{1}\{|\mathcal{A}_t| > 1\}$.
5:     **if** $Z_t = 1$ **then**
6:         Set $w_t \leftarrow \sup_{a,b \in \mathcal{A}_t} \sup_{f,f' \in \mathcal{F}_t} f(x_t, a, b) - f'(x_t, a, b)$
7:         Set $\lambda_t \leftarrow \mathbb{1}\{\sum_{s=1}^{t-1} Z_s w_s \geq \sqrt{AT/\beta}\}$.
8:         **if** $\lambda_t = 0$ **then**
9:             $p_t \leftarrow \text{Uniform}(\mathcal{A}_t)$.
10:        **else**
11:           $\gamma_t \leftarrow \sqrt{AT/\beta}$.
12:           Let $p_t$ be a solution of $\max_{a \in \mathcal{A}_t} \sum_b f_t(x_t, a, b)p_t(b) + \frac{2}{\gamma_t p_t(a)} \leq \frac{5A}{\gamma_t}$.
13:        **end if**
14:        Learner samples $a_t, b_t \sim p_t$ independently and receives the feedback $y_t$.
15:        Learner feeds $((x_t, a_t, b_t), y_t)$ to the online regression oracle which returns $f_{t+1}$.
16:     **else**
17:        Learner sets $a_t$ and $b_t$ to be the only action in $\mathcal{A}_t$, and plays them.
18:        $f_{t+1} \leftarrow f_t$.
19:     **end if**
20: **end for**

---

Next, we explain the learner's strategy for making queries. Firstly, the learner computes $w_t$, which is the "width" of the version space. Specifically, $w_t$ overestimates the instantaneous regret for playing any arm in $\mathcal{A}_t$ (Lemma 8). Then, the learner defines $\lambda_t$ that indicates if the estimated cumulative regret $\sum_{s=1}^{t-1} Z_s w_s$ has exceeded $\sqrt{AT/\beta}$. Note that $Z_t$ is multiplied to $w_t$ since no regret is incurred when $Z_t = 0$. The learner then chooses the actions (to be queried) depending on the values of $\lambda_t$:

- If $\lambda_t = 0$, the cumulative reward has not yet exceeded $\sqrt{AT/\beta} = O(\sqrt{T})$, so the learner will explore as much as possible by uniform sampling from $\mathcal{A}_t$.

- If $\lambda_t = 1$, the regret may have reached $O(\sqrt{T})$, and therefore the learner uses a technique similar to inverse gap weighting (IGW), as inspired by Saha and Krishnamurthy [2022], to achieve a better balance between exploration and exploitation. Specifically, the learner solves the convex program[4] in Line 12, which is feasible and whose solution $p_t$ satisfies (Lemma 11)

$$\mathbb{E}_{a \sim p_t}\left[ f^\star(x_t, \pi_{f^\star}(x), a) \right] = O\left( \gamma_t \mathop{\mathbb{E}}_{a,b \sim p_t}\left[ \left( f_t(x_t, a, b) - f^\star(x_t, a, b) \right)^2 \right] + \frac{A}{\gamma_t} \right). \quad (1)$$

As a result of the above relation, we can convert the instantaneous regret to the point-wise error between the predictor $f_t$ and the truth $f^\star$ plus an additive $A/\gamma_t$. This allows us to bound the cumulative point-wise error by the regret of the online regression oracle. In the special case where there exists a "reward function" $r : \mathcal{X} \times \mathcal{A} \rightarrow [0,1]$ for each $f \in \mathcal{F}$ such that $f(x, a, b) = r(x, a) - r(x, b)$ (Example 1), the solution $p_t$ can be directly written as

$$p_t(a) = \begin{cases} \frac{1}{A + \gamma_t \left( r_t(x_t, \pi_{f_t}(x_t)) - r_t(x_t, a) \right)} & a \neq \pi_{f_t}(x_t) \\ 1 - \sum_{a' \neq \pi_{f_t}(x_t)} p_t(a') & a = \pi_{f_t}(x_t) \end{cases},$$

where $r_t$ is the reward function associated with $f_t$, i.e., $f_t(x, a, b) = r_t(x, a) - r_t(x, b)$. This is the standard IGW exploration strategy [Foster and Rakhlin, 2020] and leads to the same guarantee as (1) (see Lemma 12).

---

[4]It is convex as it can be written as $|\mathcal{A}_t|$ convex constraints: $\sum_b f_t(x_t, a, b)p_t(b) + \frac{2}{\gamma_t p_t(a)} \leq \frac{5A}{\gamma_t}, \forall a \in \mathcal{A}_t$.

We discuss the computational tractability of Algorithm 1 in Appendix A. In short, it is computationally tractable for some structured function classes (e.g. linear and tabular function classes). For general function classes, it is also efficient given a regression oracle.

## 3.1 Theoretical Analysis

Towards the statistical guarantees of Algorithm 1, we employ two quantities to characterize a contextual bandit instance: the uniform gap and the eluder dimension, which are introduced below.

**Assumption 3** (Uniform gap). *We assume the optimal arm $\pi_{f^\star}(x)$ induced by $f^\star$ under any context $x \in \mathcal{X}$ is unique. Further, we assume a uniform gap $\Delta := \inf_x \inf_{a \neq \pi_{f^\star}(x)} f^\star(x, \pi_{f^\star}(x), a) > 0$.*

We note that the existence of a uniform gap is a standard assumption in the literature of contextual bandits [Dani et al., 2008, Abbasi-Yadkori et al., 2011, Audibert et al., 2010, Garivier et al., 2019, Foster and Rakhlin, 2020, Foster et al., 2021]. Next, we introduce the eluder dimension [Russo and Van Roy, 2013] and begin by defining "$\epsilon$-dependence".

**Definition 1** ($\epsilon$-dependence). *Let $\mathcal{G} \subseteq \mathcal{X} \to \mathbb{R}$ be any function class. We say an element $x \in \mathcal{X}$ is $\epsilon$-dependent on $\{x_1, x_2, \ldots, x_n\} \subseteq \mathcal{X}$ with respect to $\mathcal{G}$ if any pair of functions $g, g' \in \mathcal{G}$ satisfying $\sum_{i=1}^n (g(x_i) - g'(x_i))^2 \leq \epsilon^2$ also satisfies $g(x) - g'(x) \leq \epsilon$. Otherwise, we say $x$ is $\epsilon$-independent of $\{x_1, x_2, \ldots, x_n\}$.*

**Definition 2** (Eluder dimension). *The $\epsilon$-eluder dimension of a function class $\mathcal{G} \subseteq \mathcal{X} \to \mathbb{R}$, denoted by $\dim_E(\mathcal{G}, \epsilon)$, is the length $d$ of the longest sequence of elements in $\mathcal{X}$ satisfying that there exists some $\epsilon' \geq \epsilon$ such that every element in the sequence is $\epsilon'$-independent of its predecessors.*

Eluder dimension is a complexity measure for function classes and has been used in the literature of bandits and RL extensively [Chen et al., 2022, Osband and Van Roy, 2014, Wang et al., 2020, Foster et al., 2021, Wen and Van Roy, 2013, Jain et al., 2015, Ayoub et al., 2020, Ishfaq et al., 2021, Huang et al., 2022]. Examples where the eluder dimension is small include linear functions, generalized linear models, and functions in Reproducing Kernel Hilbert Space (RKHS).

Given these quantities, we are ready to state our main results. The proofs are provided in Appendix C.

**Theorem 1.** *Under Assumptions 1 to 3, Algorithm 1 guarantees the following upper bounds of the regret and the number of queries:*

$$\text{Regret}_T^{\text{CB}} = \widetilde{O}\left(\min\left\{\sqrt{AT\beta}, \frac{A^2 \beta^2 \dim_E(\mathcal{F}, \Delta)}{\Delta}\right\}\right),$$

$$\text{Queries}_T^{\text{CB}} = \widetilde{O}\left(\min\left\{T, \frac{A^3 \beta^3 \dim_E^2(\mathcal{F}, \Delta)}{\Delta^2}\right\}\right)$$

*with probability at least $1 - \delta$. We recall that $\beta = O(\alpha^{-1}\Upsilon + \alpha^{-2}\log(\delta^{-1}\log(T)))$, and $\alpha$ represents the coefficient of strong convexity of $\Phi$. Logarithmic terms are hidden in the upper bounds for brevity.*

For commonly used loss function (Examples 2 and 3), $\beta$ only scales logarithmically with the size (or effective size) of $\mathcal{F}$ and is thus mild. For instance, for a finite class $\mathcal{F}$, $\beta$ depends on $\log|\mathcal{F}|$. Furthermore, when $\mathcal{F}$ is infinite, we can replace $|\mathcal{F}|$ by the covering number of $\mathcal{F}$ following the standard techniques: for $d$-dimensional linear function class, $\beta$ will have a dependence of $O(d)$ (effective complexity of $\mathcal{F}$), and for the tabular class, $\beta$ will have a dependence of $O(SA^2)$. In other words, the rate of $\beta$ is acceptable as long as the complexity of the function class $\mathcal{F}$ is acceptable.

**Best-of-both-worlds guarantee.** We observe that both the regret bound and the query complexity bound consist of two components: the worst-case bound and the instance-dependent bound. The worst-case bound provides a guarantee under all circumstances, while the instance-dependent one may significantly improve the upper bound when the underlying problem is well-behaved (i.e., has a small eluder dimension and a large gap).

**Lower bounds.** To see whether these upper bounds are tight, we provide a lower bound which follows from a reduction from regular multi-armed bandits to contextual dueling bandits.

**Theorem 2** (Lower bounds). *The following two claims hold: (1) For any algorithm, there exists an instance that leads to $\text{Regret}_T^{\text{CB}} = \Omega(\sqrt{AT})$; (2) For any algorithm achieving a worse-case expected regret upper bound in the form of $\mathbb{E}[\text{Regret}_T^{\text{CB}}] = O(\sqrt{AT})$, there exists an instance with gap $\Delta = \sqrt{A/T}$ that results in $\mathbb{E}[\text{Regret}_T^{\text{CB}}] = \Omega(A/\Delta)$ and $\mathbb{E}[\text{Queries}_T^{\text{CB}}] = \Omega(A/\Delta^2) = \Omega(T)$.*

By relating these lower bounds to Theorem 1, we conclude that our algorithm achieves a tight dependence on the gap $\Delta$ and the number of rounds $T$ up to logarithmic factors. Furthermore, as an additional contribution, we establish an alternative lower bound in Section C.4.1 by conditioning on the limit of regret, rather than the worst-case regret as assumed in Theorem 2.

**Intuition of proofs.** We next provide intuition for why our algorithm has the aforementioned theoretical guarantees. First, we observe that from the definition of $\lambda_t$, the left term inside the indicator is non-decreasing, which allows us to divide rounds into two phases. In the first phase, $\lambda_t$ is always 0, and then at some point, it changes to 1 and remains 1 for the rest rounds. After realizing this, we first explain the intuition of the worst-case regret. In the first phase, as $w_t$ is an overestimate of the instantaneous regret (see Lemma 8), the accumulated regret in this phase cannot exceed $O(\sqrt{T})$. In the second phase, we adapt the analysis of IGW to this scenario to obtain an $O(\sqrt{T})$ upper bound. A similar technique has been used in Saha and Krishnamurthy [2022], Foster et al. [2021]. As the regret in both phases is at most $O(\sqrt{T})$, the total regret cannot exceed $O(\sqrt{T})$. Next, we explain the intuition of instance-dependent regret. Due to the existence of a uniform gap $\Delta$, we can first prove that as long as $|\mathcal{A}_t| > 1$, we must have $w_t \geq \Delta$ (see Lemma 7). This means that for all rounds that may incur regret, the corresponding width is at least $\Delta$. However, this cannot happen too many times as this frequency is bounded by the eluder dimension, which leads to an instance-dependent regret upper bound. Leveraging a similar technique, we can also obtain an upper bound on the number of queries.

**Comparison to regret bounds of dueling bandits.** As established by prior works [Yue et al., 2012, Saha and Gaillard, 2022], for dueling bandits, the minimax regret rate is $\tilde{\Theta}(\sqrt{AT})$ and the instance-dependent rate is $\tilde{\Theta}(A/\Delta)$. If we reduce our result (Theorem 1) into the dueling bandits setting, we will get

$$\mathrm{Regret}_T = \tilde{O}\left(\min\left\{\sqrt{AT}, \frac{A^2 \mathrm{dim}_E(\mathcal{F}, \frac{\Delta}{2A^2})}{\Delta}\right\}\right) = \tilde{O}\left(\min\left\{\sqrt{AT}, \frac{A^3}{\Delta}\right\}\right)$$

where the second equality holds since the eluder dimension is upper bounded by $A$ for dueling bandits. We observe that the worst-case regret rate is the same, but there is a gap of $A^2$ in the instance-dependent bound. The improvement of this gap is an interesting future direction.

**Comparion to MINMAXDB [Saha and Krishnamurthy, 2022].** In this prior work, the authors assume that $\Pr(y = 1 \mid x, a, b) = (f^\star(x, a, b) + 1)/2$, which is a specification of our feedback model (Example 2). While our worst-case regret bound matches their regret bound, our paper improves upon their results by having an additional instance-dependent regret bound that depends on the eluder dimension and gap. Furthermore, we also provide bounds on the query complexity which could be small for benign instances while MINMAXDB simply queries on every round.

**Comparison to ADACB [Foster et al., 2021].** Our method shares some similarities with Foster et al. [2021], especially in terms of theoretical results, but differs in two aspects: (1) they assume regular contextual bandits where the learner observes the reward directly, while we assume preference feedback, and (2) they assume a stochastic setting where contexts are drawn i.i.d., but we assume that the context is adversarially chosen. While these two settings may not be directly comparable, it should be noted that [Foster et al., 2021] do not aim to minimize query complexity.

**Results without the uniform gap assumption.** We highlight that Theorem 1 can naturally extend to scenarios where a uniform gap does not exist (i.e., when Assumption 3 is not satisfied) without any modifications to the algorithm. The result is stated below, which is analogous to Theorem 1.

**Theorem 3.** *Under Assumptions 1 and 2, Algorithm 1 guarantees the following upper bounds of the regret and the number of queries:*

$$\mathrm{Regret}_T^{\mathrm{CB}} = \widetilde{O}\left(\min\left\{\sqrt{ATB}, \min_{\epsilon > 0}\left\{T_\epsilon\beta + \frac{A^2\beta^2 \mathrm{dim}_E(\mathcal{F}, \epsilon)}{\epsilon}\right\}\right\}\right),$$

$$\mathrm{Queries}_T^{\mathrm{CB}} = \widetilde{O}\left(\min\left\{T, \min_{\epsilon > 0}\left\{T_\epsilon^2\beta/A + \frac{A^3\beta^3 \mathrm{dim}_E^2(\mathcal{F}, \epsilon)}{\epsilon^2}\right\}\right\}\right)$$

*with probability at least $1 - \delta$. Here we define the gap of context $x$ as $\mathrm{Gap}(x) := \min_{a \neq \pi_{f^\star}(x)} f^\star(x, \pi_{f^\star}(x), a)$ and the number of rounds where contexts have small gap as $T_\epsilon := \sum_{t=1}^T \mathbb{1}\{\mathrm{Gap}(x_t) \leq \epsilon\}$. We also recall that $\beta = O(\alpha^{-1}\Upsilon + \alpha^{-2}\log(\delta^{-1}\log(T)))$, and $\alpha$ denotes the coefficient of strong convexity of $\Phi$.*

---

**Algorithm 2** Active preference qUeRy fOR imitAtion lEarning (AURORAE)

---

**Require:** Function class $\mathcal{F}_0, \mathcal{F}_1, \ldots, \mathcal{F}_{H-1}$, confidence parameter $\beta$.
 1: Learner creates $H$ instances of Algorithm 1: $\text{AURORA}_h(\mathcal{F}_h, \beta)$ for $h = 0, 1, \ldots, H-1$.
 2: **for** $t = 1, 2, \ldots, T$ **do**
 3:     Learner receive initial state $x_{t,0}$.
 4:     **for** $h = 0, 1, \ldots, H-1$ **do**
 5:         Learner feeds $x_{t,h}$ to $\text{AURORA}_h(\mathcal{F}_h, \beta)$, and receives back $a_{t,h}, b_{t,h}, Z_{t,h}$.
 6:         **if** $Z_{t,h} = 1$ **then**
 7:             Learner receives feedback $y_{t,h}$.
 8:             Learner feeds $((x_{t,h}, a_{t,h}, b_{t,h}), y_{t,h})$ to $\text{AURORA}_h(\mathcal{F}_h, \beta)$ to update its online regression oracle and local variables.
 9:         **end if**
10:         Learner executes $a \sim \text{Uniform}(\{a_{t,h}, b_{t,h}\})$ and transits to $x_{t,h+1}$.
11:     **end for**
12: **end for**

---

Compared to Theorem 1, the above result has an extra gap-dependent term, $T_\epsilon$, measuring how many times the context falls into a small-gap region. We highlight that $T_\epsilon$ is small under certain conditions such as the Tsybakov noise condition [Tsybakov, 2004]. It is also worth mentioning that our algorithm is agnostic to $\epsilon$, thus allowing us to take the minimum over all $\epsilon > 0$.

**Comparion to SAGE-BANDIT [Sekhari et al., 2023].** Theorem 3 is similar to Theorem 4 in Sekhari et al. [2023], which studies active queries in contextual bandits with standard reward signal. Although our result looks slightly worse in terms of the factor $A$, we believe that this inferiority is reasonable since our approach requires two actions to form a query, thus analytically expanding the action space to $\mathcal{A}^2$. Whether this dependency can be improved remains a question for future investigation.

## 4 Imitation Learning with Preference-Based Active Queries

In this section, we introduce our second algorithm, which is presented in Algorithm 2 for imitation learning. In essence, the learner treats the MDP as a concatenation of $H$ contextual bandits and runs an instance of AURORA (Algorithm 1) for each time step. Specifically, the learner first creates $H$ instances of AURORA, denoted by $\text{AURORA}_h$ (for $h = 0, \ldots, H-1$). Here, $\text{AURORA}_h$ should be thought of as an interactive program that takes the context $x$ as input and outputs $a$, $b$, and $Z$. At each episode $t$, and each step $h$ therein, the learner first feeds the current state $x_{t,h}$ to $\text{AURORA}_h$ as the context; then, $\text{AURORA}_h$ decides whether to query (i.e. $Z_{t,h}$) and returns the actions $a_{t,h}$ and $b_{t,h}$. If it decides to make a query, the learner will ask for the feedback $y_{t,h}$ on the proposed actions $a_{t,h}, b_{t,h}$, and provide the information $((x_{t,h}, a_{t,h}, b_{t,h}), y_{t,h})$ back to $\text{AURORA}_h$ to update its online regression oracle (and other local variables). We recall that the noisy binary feedback $y_{t,h}$ is sampled as $y_{t,h} \sim \phi(Q_h^{\pi_e}(x_{t,h}, a_{t,h}) - Q_h^{\pi_e}(x_{t,h}, b_{t,h}))$, and also emphasize that the learner neither has access to $a \sim \pi_e(x_{t,h})$ like in DAGGER [Ross et al., 2011] nor reward-to-go like in AGGREVATE(D) [Ross and Bagnell, 2014, Sun et al., 2017]. Finally, the learner chooses one of the two actions uniformly at random, executes it in the underlying MDP, and transits to the next state $x_{t,h+1}$ in the episode. The above process is then repeated with $\text{AURORA}_{h+1}$ till the episode ends. We name this algorithm AURORAE, the plural form of AURORA, which signifies that the algorithm is essentially a stack of multiple AURORA instances.

### 4.1 Theoretical Analysis

As Algorithm 2 is essentially a stack of Algorithm 1, we can inherit many of the theoretical guarantees from the previous section. To state the results, we first extend Assumption 3 into imitation learning.

**Assumption 4** (Uniform Gap). *Let $f_h^\star$ be defined such that for any $x \in \mathcal{X}$, $a, b \in \mathcal{A}^2$, $f_h^\star(x, a, b) = Q_h^{\pi_e}(x, a) - Q_h^{\pi_e}(x, b)$. For all $h$, we assume the optimal action for $f_h^\star$ under any state $x \in \mathcal{X}$ is unique. Further, we assume a uniform gap $\Delta := \inf_h \inf_x \inf_{a \neq \pi_{f_h^\star}(x)} f_h^\star(x, \pi_{f_h^\star}(x), a) > 0$.*

This assumption essentially says that $Q_h^{\pi_e}$ has a gap in actions. We remark that, just as Assumption 3 is a common condition in the bandit literature, Assumption 4 is also common in MDPs [Du et al.,

2019, Foster et al., 2021, Simchowitz and Jamieson, 2019, Jin and Luo, 2020, Lykouris et al., 2021, He et al., 2021]. The theoretical guarantee for Algorithm 2 is presented in Theorem 4. We note a technical difference between this result and Theorem 1: although we treat the MDP as a concatenation of $H$ contextual bandits, the instantaneous regret of imitation learning is defined as the performance gap between the combined policy $\pi_t$ derived from the $H$ instances as a cohesive unit and the expert policy. This necessitates the use of performance difference lemma (Lemma 5) to get a unified result.

**Theorem 4.** *Under Assumptions 1, 2 and 4, Algorithm 2 guarantees the following upper bounds of the regret and the number of queries:*

$$\text{Regret}_T^{\text{IL}} \leq \widetilde{O}\left(H \cdot \min\left\{\sqrt{AT\beta}, \frac{A^2\beta^2\text{dim}_E\left(\mathcal{F}, \Delta\right)}{\Delta}\right\}\right) - \text{Adv}_T,$$

$$\text{Queries}_T^{\text{IL}} \leq \widetilde{O}\left(H \cdot \min\left\{T, \frac{A^3\beta^3\text{dim}_E^2\left(\mathcal{F}, \Delta\right)}{\Delta^2}\right\}\right)$$

*with probability at least $1 - \delta$. Here $\text{Adv}_T := \sum_{t=1}^{T}\sum_{h=0}^{H-1}\mathbb{E}_{x_{t,h}\sim d_{x_{t,0},h}^{\pi_t}}[\max_a A_h^{\pi_e}(x_{t,h}, a)]$ is non-negative, and $d_{x_{t,0},h}^{\pi_t}(x)$ denotes the probability of $\pi_t$ [5] reaching the state $x$ at time step $h$ starting from inital state $x_{t,0}$. In the above, $\beta = O(\alpha^{-1}\Upsilon + \alpha^{-2}\log(H\delta^{-1}\log(T)))$ and $\alpha$ denotes the coefficient of strong convexity of $\Phi$.*

Compared to Theorem 1, the main terms of the upper bounds for imitation learning are precisely the bounds in Theorem 1 multiplied by $H$. In the proof presented in Appendix C.6, we use the performance difference lemma to reduce the regret of imitation learning to the sum of the regret of $H$ contextual dueling bandits, which explains this additional factor of $H$.

Another interesting point is that the main term of the regret upper bound is subtracted by a non-negative term $\text{Adv}_T$, which measures the degree to which we can *outperform* the expert policy. In other words, our algorithm not only competes with the expert policy but can also surpass it to some extent. To see this, let us consider the *average regret*, which is defined as $\text{AveRegret}_T^{\text{IL}} := \text{Regret}_T^{\text{IL}}/T = \sum_{t=1}^{T}(V_0^{\pi_e}(x_{t,0}) - V_0^{\pi_t}(x_{t,0}))/T$. Then, Theorem 4 implies that $\text{AveRegret}_T^{\text{IL}} = O(H\sqrt{A\beta/T}) - \text{Adv}_T/T$ where we have simplified it by ignoring the instance-dependent upper bound and logarithmic factors for clarity. Now, consider a case where $\max_a A_h^{\pi_e}(x, a) > \alpha_0$ for some constant $\alpha_0 > 0$ for all $x$ and $h$. This can happen when the expert policy is suboptimal for every state. Consequently, we have $\text{Adv}_T > \alpha_0 HT$. In this case, the average regret is further bounded by $\text{AveRegret}_T^{\text{IL}} = O(H\sqrt{A\beta/T}) - \alpha_0 H$. When $T \to \infty$, we have $\text{AveRegret}_T^{\text{IL}} \to -\alpha_0 H < 0$. This means that the best (or average) learned policy will eventually outperform the expert policy. This guarantee is stronger than that of DAGGER [Ross et al., 2011] in that DAGGER cannot ensure the learned policy is better than the expert policy regardless of how suboptimal the expert may be. While this may look surprising at first glance since we are operating under a somewhat weaker query mode than that of DAGGER, we note that by querying experts for comparisons on pairs of actions with feedback sampling as $y \sim \phi(Q^{\pi_e}(x, a) - Q^{\pi_e}(x, b))$, it is possible to identify the action that maximizes $Q^{\pi_e}(x, a)$ (even if we cannot identify the value $Q^{\pi_e}(x, a)$). Finally, we remark that our worst-case regret bound is similar to that of Ross and Bagnell [2014], Sun et al. [2017], which can also outperform a suboptimal expert but require access to both expert's actions and reward signals — a much stronger query model than ours.

## 5    Conclusion

We presented interactive decision-making algorithms that learn from preference-based feedback while minimizing query complexity. Our algorithms for contextual bandits and imitation learning share worst-case regret bounds similar to the bounds of the state-of-the-art algorithms in standard settings while maintaining instance-dependent regret bounds and query complexity bounds. Notably, our imitation learning algorithm can outperform suboptimal experts, matching the result of Ross and Bagnell [2014], Sun et al. [2017], which operates under much stronger feedback.

---

[5]Policy $\pi_t$ consists of $H$ time-dependent policies $\pi_{t,1}, \ldots, \pi_{t,H}$, where each $\pi_{t,h}$ is defined implicitly via AURORA$_h$, i.e., $\pi_{t,h}$ generates action as follows: given $x_{t,h}$, AURORA$_h$ recommends $a_{t,h}, b_{t,h}$, followed by uniformly sampling an action from $\{a_{t,h}, b_{t,h}\}$.

## Acknowledgements

AS acknowledges support from the Simons Foundation and NSF through award DMS-2031883, as well as from the DOE through award DE-SC0022199. KS acknowledges support from NSF CAREER Award 1750575, and LinkedIn-Cornell grant.

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

# A  Computational tractability of Algorithm 1

We observe that the computational complexity of the proposed algorithm mainly depends on the computation of the candidate arm set (Line 3) and the width (Line 6). When $\mathcal{F}$ is a $d$-dimensional linear function class, the computational complexity can be $\tilde{O}(dTA)$ since the version space exhibits an ellipsoid structure and thus both the candidate arm and the width can be computed in $\tilde{O}(dA)$ time. When $\mathcal{F}$ is tabular, it can be considered as a special case of linear class with one-hot encoding. In this case, we have $d = S \times A$, resulting in a computational complexity of $\tilde{O}(TSA^2)$.

For a more general convex function class $\mathcal{F}$, we can design an efficient algorithm based on a weighted regression oracle for $\mathcal{F}$. To this end, we first note that an approach to efficiently compute the width has been proposed by Foster et al. [2018a], and now we propose the following method to compute the candidate arm set. An arm $a$ belongs to the candidate arm set at round $t$ if and only if the following optimization problem (with an constant objective) is feasible

$$\min_{f \in \mathcal{F}, \xi \in \mathbb{R}^A} 1 \quad \text{s.t.} \quad f(x, a, a') = \xi_{a'}, \quad \xi_{a'} > 0 \quad (\forall a' \neq a),$$

$$\text{and} \quad \sum_{s=1}^{t-1} Z_s \left( f(x_s, a_s, b_s) - f_t(x_s, a_s, b_s) \right)^2 \leq \beta.$$

Here we introduce the slack variable $\xi$ so that the optimization part for $f$ can be simply reduced to a weighted regression oracle. Next, we convert the above into Lagrangian formulation and obtain

$$\min_{f \in \mathcal{F}, \xi \in \mathbb{R}^A} \max_{\alpha \in \mathbb{R}_+^A, \gamma \in \mathbb{R}_+^A, \lambda \in \mathbb{R}_+} 1 + \sum_{a' \neq a} \alpha_{a'} \left( f(x, a, a') - \xi_{a'} \right)^2 - \sum_{a' \neq a} \gamma_{a'} \xi_{a'}$$

$$+ \lambda \left( \sum_{s=1}^{t-1} Z_s \left( f(x_s, a_s, b_s) - f_t(x_s, a_s, b_s) \right)^2 - \beta \right)$$

$$= \max_{\alpha \in \mathbb{R}_+^A, \gamma \in \mathbb{R}_+^A, \lambda \in \mathbb{R}_+} \min_{f \in \mathcal{F}, \xi \in \mathbb{R}^A} 1 + \sum_{a' \neq a} \alpha_{a'} \left( f(x, a, a') - \xi_{a'} \right)^2 - \sum_{a' \neq a} \gamma_{a'} \xi_{a'}$$

$$+ \lambda \left( \sum_{s=1}^{t-1} Z_s \left( f(x_s, a_s, b_s) - f_t(x_s, a_s, b_s) \right)^2 - \beta \right).$$

Here we can swap the min and max since the objective is convex in the joint space of $f$ and $\xi$. Then, the inner minimization problem can be solved by updating $f$ via the regression oracle and updating $\xi$ via gradient descent; for the outer maximization problem, we can do projected gradient ascent.

# B  Preliminaries

**Lemma 2** (Kakade and Tewari [2008, Lemma 3]). *Suppose $X_1, \ldots, X_T$ is a martingale difference sequence with $|X_t| \leq b$. Let*

$$\text{Var}_t X_t = \text{Var}(X_t \mid X_1, \ldots, X_{t-1})$$

*Let $V = \sum_{t=1}^T \text{Var}_t X_t$ be the sum of conditional variances of $X_t$'s. Further, let $\sigma = \sqrt{V}$. Then we have, for any $\delta < 1/e$ and $T \geq 3$,*

$$\Pr\left( \sum_{t=1}^T X_t > \max\{2\sigma, 3b\sqrt{\ln(1/\delta)}\} \sqrt{\ln(1/\delta)} \right) \leq 4 \ln(T)\delta.$$

**Lemma 3** (Foster and Rakhlin [2020, Lemma 3]). *For any vector $\hat{y} \in [0, 1]^A$, if we define $p$ to be*

$$p(a) = \begin{cases} \frac{1}{A + \gamma \left( \hat{y}(\hat{a}) - \hat{y}(a) \right)} & \text{if } a \neq \hat{a}, \\ 1 - \sum_{a \neq \hat{a}} p(a) & \text{if } a = \hat{a} \end{cases}$$

*where $\hat{a} = \arg\max_a \hat{y}(a)$, then for any $y^\star \in [0, 1]^A$ and $\gamma > 0$, we have*

$$\mathbb{E}_{a \sim p} \left[ \left( y^\star(a^\star) - y^\star(a) \right) - \gamma \left( \hat{y}(a) - y^\star(a) \right)^2 \right] \leq \frac{A}{\gamma}.$$

**Lemma 4** ([Zhu and Nowak [2022], Lemma 2]). *Let $(Z_t)_{t \leq T}$ to be real-valued sequence of positive random variables adapted to a filtration $\mathfrak{F}_t$. If $|Z_t| \leq B$ almost surely, then with probability at least $1 - \delta$,*

$$\sum_{t=1}^{T} Z_t \leq \frac{3}{2} \sum_{t=1}^{T} \mathbb{E}_t [Z_t] + 4B \log \left(2\delta^{-1}\right),$$

*and*

$$\sum_{t=1}^{T} \mathbb{E}_t [Z_t] \leq 2 \sum_{t=1}^{T} Z_t + 8B \log \left(2\delta^{-1}\right).$$

**Lemma 5** (Performance difference lemma [Agarwal et al., 2019]). *For any two policies $\pi$ and $\pi'$ and any state $x_0 \in \mathcal{X}$, we have*

$$V_0^{\pi}(x_0) - V_0^{\pi'}(x_0) = \sum_{h=0}^{H-1} \mathbb{E}_{x_h, a_h \sim d_{x_0, h}^{\pi}} \left[A_h^{\pi'}(x_h, a_h)\right]$$

*where $A_h^{\pi}(x, a) = Q_h^{\pi}(x, a) - V_h^{\pi}(x, a)$ and $d_{x_0, h}^{\pi}(x, a)$ is the probability of $\pi$ reaching the state-action pair $(x, a)$ at time step $h$ starting from initial state $x_0$.*

**Lemma 6.** *For any two Bernoulli distributions $\mathrm{Bern}(x)$ and $\mathrm{Bern}(y)$ with $x, y \in [b, 1 - b]$ for some $0 < b \leq 1/2$, the KL divergence is bounded as*

$$\mathrm{KL}\Big(\mathrm{Bern}(x), \mathrm{Bern}(y)\Big) \leq \frac{2(x - y)^2}{b}.$$

*Proof of Lemma 6.* Denote $\Delta = x - y$. Then, by definition, we have

$$
\begin{aligned}
\mathrm{KL}\Big(\mathrm{Bern}(x), \mathrm{Bern}(y)\Big) =& x \ln \frac{x}{y} + (1 - x) \ln \frac{1 - x}{1 - y} \\
=& x \ln \frac{x}{x - \Delta} + (1 - x) \ln \frac{1 - x}{1 - x + \Delta} \\
=& x \ln \left(1 + \frac{\Delta}{x - \Delta}\right) + (1 - x) \ln \left(1 - \frac{\Delta}{1 - x + \Delta}\right)
\end{aligned}
$$

Since $\ln(1 + x) \leq x$ for all $x > -1$, we have

$$
\begin{aligned}
\mathrm{KL}\Big(\mathrm{Bern}(x), \mathrm{Bern}(y)\Big) \leq& x \cdot \frac{\Delta}{x - \Delta} - (1 - x) \cdot \frac{\Delta}{1 - x + \Delta} \\
=& \Delta \cdot \left(\frac{x}{x - \Delta} - \frac{1 - x}{1 - x + \Delta}\right) \\
=& \Delta \cdot \left(\frac{\Delta}{x - \Delta} + \frac{\Delta}{1 - x + \Delta}\right) \\
\leq& \Delta^2 \cdot \left(\frac{1}{y} + \frac{1}{1 - y}\right) \leq \frac{2\Delta^2}{b}.
\end{aligned}
$$

$\square$

## C  Missing Proofs

### C.1  Supporting Lemmas

**Definition 3** (Strong convexity). *A function $\Phi : [-1, 1] \to \mathbb{R}$ is $\alpha$-strongly-convex if for all $u, u' \in \mathbb{R}$, we have*

$$\frac{\alpha}{2}(u' - u)^2 \leq \Phi(u') - \Phi(u) - \nabla\Phi(u)(u' - u).$$

*where $\nabla\Phi$ means the derivative of $\Phi$.*

**Lemma 7.** *For any $t \in [T]$, if $f^{\star} \in \mathcal{F}_t$, then we have $w_t \geq \Delta$ whenever $|\mathcal{A}_t| > 1$.*

*Proof of lemma 7.* When $|\mathcal{A}_t| > 1$, we know there exists a function $f' \in \mathcal{F}_t$ satisfying
$$a' := \pi_{f'}(x_t) \neq \pi_{f^\star}(x_t) =: a_t^\star.$$
Then we have $\Delta \leq f^\star(x_t, a_t^\star, a') \leq f^\star(x_t, a_t^\star, a') - f'(x_t, a_t^\star, a') \leq w_t$ where the second inequality holds since $f'(x_t, a_t^\star, a') \leq 0$. $\qquad\square$

**Lemma 8.** *For any $t \in [T]$ and any arm $a \in \mathcal{A}_t$, we have $f^\star(x_t, \pi_{f^\star}(x_t), a) \leq w_t$.*

*Proof of Lemma 8.* For any $a \in \mathcal{A}_t$, by the definition of $\mathcal{A}_t$, there must exists a function $f$ for which $a = \pi_f(x_t)$. Hence,
$$f^\star(x_t, \pi_{f^\star}(x_t), a) \leq f^\star(x_t, \pi_{f^\star}(x_t), a) - f(x_t, \pi_{f^\star}(x_t), a) \leq w_t,$$
where the first inequality holds since $f(x_t, \pi_{f^\star}(x_t), a) \leq 0$. $\qquad\square$

The following lemma is adapted from Agarwal [2013, Lemma 2].

**Lemma 9.** *The following holds with probability at least $1 - \delta$ for any $T > 3$,*
$$\sum_{t=1}^{T} Z_t \big(f^\star(x_t, a_t, b_t) - f_t(x_t, a_t, b_t)\big)^2 \leq \frac{4\Upsilon}{\alpha} + \frac{16 + 24\alpha}{\alpha^2} \log\big(4\delta^{-1}\log(T)\big).$$

*Proof of Lemma 9.* Throughout the proof, we denote $z_t := (x_t, a_t, b_t)$ for notational simplicity. We define $D_\Phi$ as the Bregman divergence of the function $\Phi$:
$$D_\Phi(u, v) = \Phi(u) - \Phi(v) - \phi(v)(u - v)$$
where we recall that $\phi = \Phi'$ is the derivative of $\Phi$. Since $\Phi$ is $\alpha$-strong convex, we have $\alpha(u-v)^2/2 \leq D_\Phi(u, v)$, and hence,
$$\sum_{t=1}^{T} Z_t \big(f^\star(z_t) - f_t(z_t)\big)^2 \leq \frac{2}{\alpha} \sum_{t=1}^{T} Z_t D_\Phi(f_t(z_t), f^\star(z_t)). \tag{2}$$
Hence, it suffice to derive an upper bound for the Bregman divergence in the right hand side above. Define $\nu_t$ as below:
$$
\begin{aligned}
\nu_t :=& Z_t \Big[ D_\Phi\left(f_t(z_t), f^\star(z_t)\right) - \left(\ell_\phi\left(f_t(z_t), y_t\right) - \ell_\phi\left(f^\star(z_t), y_t\right)\right) \Big] \\
=& Z_t \Big[ D_\Phi\left(f_t(z_t), f^\star(z_t)\right) - \left(\Phi\left(f_t(z_t)\right) - (y_t + 1)f_t(z_t)/2 - \Phi\left(f^\star(z_t)\right) + (y_t + 1)f^\star(z_t)/2\right) \Big] \\
=& Z_t \Big[ \Phi\left(f_t(z_t)\right) - \Phi\left(f^\star(z_t)\right) - \phi\left(f^\star(z_t)\right)\left(f_t(z_t) - f^\star(z_t)\right) \\
& - \left(\Phi\left(f_t(z_t)\right) - (y_t + 1)f_t(z_t)/2 - \Phi\left(f^\star(z_t)\right) + (y_t + 1)f^\star(z_t)/2\right) \Big] \\
=& Z_t \big(f_t(z_t) - f^\star(z_t)\big)\big((y_t + 1)/2 - \phi(f^\star(z_t))\big)
\end{aligned}
$$
We note that $\mathbb{E}_t[(y_t + 1)/2] = \phi(f^\star(z_t))$, and thus $\mathbb{E}_t[\nu_t] = 0$, which means $\nu_t$ is a martingale difference sequence. Now we bound the value and the conditional variance of $\nu_t$ in order to derive concentration results.

1. Bound the value of $\nu_t$:
$$|\nu_t| \leq |(y_t + 1)/2 - \phi\left(f^\star(z_t)\right)| \cdot |f_t(z_t) - f^\star(z_t)| \leq 1 \cdot 2 = 2.$$

2. Bound the conditional variance of $\nu_t$:
$$
\begin{aligned}
\mathbb{E}_t[\nu_t^2] =& Z_t \mathbb{E}_t \left[ ((y_t + 1)/2 - \phi\left(f^\star(z_t)\right))^2 \left(f_t(z_t) - f^\star(z_t)\right)^2 \right] \\
\leq& Z_t \mathbb{E}_t \left[ \left(f_t(z_t) - f^\star(z_t)\right)^2 \right] \\
\leq& Z_t \mathbb{E}_t \left[ \frac{2}{\alpha} \cdot D_\Phi(f_t(z_t), f^\star(z_t)) \right] \\
\leq& \frac{2Z_t}{\alpha} D_\Phi(f_t(z_t), f^\star(z_t))
\end{aligned}
$$
where for the last line we note that $x_t, g_t$ are measurable at $t$.

Now we apply Lemma 2, which yields for any $\delta < 1/e$ and $T > 3$, with probability at least $1 - 4\delta \log(T)$,

$$\sum_{t=1}^{T} \nu_t \leq \max\left\{ 2\sqrt{\sum_{t=1}^{T} \frac{2Z_t}{\alpha} D_\Phi(f_t(z_t), f^\star(z_t))}, 6\sqrt{\log(1/\delta)} \right\} \sqrt{\log(1/\delta)}$$

$$\leq 2\sqrt{\sum_{t=1}^{T} \frac{2Z_t}{\alpha} D_\Phi(f_t(z_t), f^\star(z_t))\log(1/\delta)} + 6\log(1/\delta) \qquad \text{(since } \max(a,b) \leq a + b\text{)}$$

$$\leq \sum_{t=1}^{T} \frac{1}{2} Z_t D_\Phi(f_t(z_t), f^\star(z_t)) + \frac{4\log(1/\delta)}{\alpha} + 6\log(1/\delta) \qquad \text{(AM-GM)}$$

Recall the definition of $\nu_t$, and we conclude that

$$\sum_{t=1}^{T} Z_t D_\Phi\left(f_t(z_t), f^\star(z_t)\right) - \sum_{t=1}^{T} Z_t\left(\ell_\phi\left(f_t(z_t), y_t\right) - \ell_\phi\left(f^\star(z_t), y_t\right)\right) \leq$$

$$\sum_{t=1}^{T} \frac{1}{2} Z_t D_\Phi(f_t(z_t), f^\star(z_t)) + \frac{4\log(1/\delta)}{\alpha} + 6\log(1/\delta),$$

which implies

$$\frac{1}{2} \sum_{t=1}^{T} Z_t D_\Phi\left(f_t(z_t), f^\star(z_t)\right) \leq \sum_{t=1}^{T} Z_t\left(\ell_\phi\left(f_t(z_t), y_t\right) - \ell_\phi\left(f^\star(z_t), y_t\right)\right) + \frac{4\log(1/\delta)}{\alpha} + 6\log(1/\delta).$$

Plugging this upper bound of Bregman divergence into (2), we obtain that, with probability at least $1 - 4\delta \log(T)$, for any $\delta < 1/e$ and $T > 3$, we have

$$\sum_{t=1}^{T} Z_t\left(f^\star(z_t) - f_t(z_t)\right)^2 \leq \frac{4}{\alpha}\Upsilon + \left(\frac{16}{\alpha^2} + \frac{24}{\alpha}\right)\log(\delta^{-1}) =: \beta$$

Finally, we finish the proof by adjusting the coefficient $\delta$ and taking a union bound to obtain the desired result. $\qquad\square$

The following lemma is a variant of Russo and Van Roy [2013, Proposition 3], with the main difference being that (1) the version space is established using the function produced by the oracle instead of the least squares estimator, and (2) the extra multiplicative factor $Z_t$.

**Lemma 10.** *For Algorithm 1, it holds that*

$$\sum_{t=1}^{T} Z_t \mathbb{1}\left\{ \sup_{f,f'\in\mathcal{F}_t} f(x_t, a_t, b_t) - f'(x_t, a_t, b_t) > \epsilon \right\} \leq \left(\frac{4\beta}{\epsilon^2} + 1\right) \dim_E(\mathcal{F}, \epsilon) \qquad (3)$$

*for any constant $\epsilon > 0$,*

*Proof of Lemma 10.* We first define a subsequence consisting only of the elements for which we made a query in that round. Specifically, we define $((x_{i_1}, a_{i_1}, b_{i_1}), (x_{i_2}, a_{i_2}, b_{i_2}), \ldots, (x_{i_k}, a_{i_k}, b_{i_k}))$ where $1 \leq i_1 < i_2 < \cdots < i_k \leq T$ and $(x_t, a_t, b_t)$ belongs to the subsequence if and only if $Z_t = 1$. We further simplify the notation by defining $z_j := (x_{i_j}, a_{i_j}, b_{i_j})$ and $f(z_j) := f(x_{i_j}, a_{i_j}, b_{i_j})$. Then we note that the left-hand side of (3) is equivalent to

$$\sum_{j=1}^{k} \mathbb{1}\left\{ \sup_{f,f'\in\mathcal{F}_j} f(z_j) - f'(z_j) > \epsilon \right\}, \qquad (4)$$

and the version space in Algorithm 1 is equal to

$$\mathcal{F}_j = \left\{ f \in \mathcal{F} : \sum_{s=1}^{j-1} \left(f(z_s) - f_s(z_s)\right)^2 \leq \beta \right\}. \qquad (5)$$

Hence, it suffice to establish the lower bound for (4) under the version space of (5). To that end, we make one more simplification in notation: we denote

$$w'_j := \sup_{f,f' \in \mathcal{F}_j} f(z_j) - f'(z_j)$$

We begin by showing that if $w'_j > \epsilon$ for some $j \in [k]$, then $z_j$ is $\epsilon$-dependent on at most $4\beta/\epsilon^2$ disjoint subsequence of its predecessors. To see this, we note that when $w'_j > \epsilon$, there must exist two function $f, f' \in \mathcal{F}_j$ such that $f(z_j) - f'(z_j) > \epsilon$. If $z_j$ is $\epsilon$-dependent on a subsequence $(z_{i_1}, z_{i_2}, \ldots, z_{i_n})$ of its predecessors, we must have

$$\sum_{s=1}^{n} \big(f(z_{i_s}) - f'(z_{i_s})\big)^2 > \epsilon^2.$$

Hence, if $z_j$ is $\epsilon$-dependent on $l$ disjoint subsequences, we have

$$\sum_{s=1}^{j-1} \big(f(z_s) - f'(z_s)\big)^2 > l\epsilon^2. \tag{6}$$

For the left-hand side above, we also have

$$\sum_{s=1}^{j-1} \big(f(z_s) - f'(z_s)\big)^2 \leq 2\sum_{s=1}^{j-1} \big(f(z_s) - f_s(z_s)\big)^2 + 2\sum_{s=1}^{j-1} \big(f_s(z_s) - f'(z_s)\big)^2 \leq 4\beta \tag{7}$$

where the first inequality holds since $(a+b)^2 \leq 2(a^2+b^2)$ for any $a, b$, and the second inequality holds by (5). Combining (6) and (7), we get that $l \leq 4\beta/\epsilon^2$.

Next, we show that for any sequence $(z'_1, \ldots, z'_\tau)$, there is at least one element that is $\epsilon$-dependent on at least $\tau/d - 1$ disjoint subsequence of its predecessors, where $d := \dim_E(\mathcal{F}, \epsilon)$. To show this, let $m$ be the integer satisfying $md + 1 \leq \tau \leq md + d$. We will construct $m$ disjoint subsequences, $B_1, \ldots, B_m$. At the beginning, let $B_i = (z'_i)$ for $i \in [m]$. If $z'_{m+1}$ is $\epsilon$-dependent on each subsequence $B_1, \ldots, B_m$, then we are done. Otherwise, we select a subsequence $B_i$ which $z'_{m+1}$ is $\epsilon$-independent of and append $z'_{m+1}$ to $B_i$. We repeat this process for all elements with indices $j > m + 1$ until either $z'_j$ is $\epsilon$-dependent on each subsequence or $j = \tau$. For the latter, we have $\sum_{i=1}^{m} |B_i| \geq md$, and since each element of a subsequence $B_i$ is $\epsilon$-independent of its predecesors, we must have $|B_i| = d$ for all $i$. Then, $z_\tau$ must be $\epsilon$-dependent on each subsequence by the definition of eluder dimension.

Finally, let's take the sequence $(z'_1, \ldots z'_\tau)$ to be the subsequence of $(z_1, \ldots, z_k)$ consisting of elements $z_j$ for which $w'_j > \epsilon$. As we have established, we have (1) each $z'_j$ is $\epsilon$-dependent on at most $4\beta/\epsilon^2$ disjoint subsequences, and (2) some $z'_j$ is $\epsilon$-dependent on at least $\tau/d - 1$ disjoint subsequences. Therefore, we must have $\tau/d - 1 \leq 4\beta/\epsilon^2$, implying that $\tau \leq (4\beta/\epsilon^2 + 1)d$. $\quad\square$

The following lemma is adopted from Saha and Krishnamurthy [2022, Lemma 3].

**Lemma 11.** *For any function $f \in \mathcal{F}$ and any context $x \in \mathcal{X}$, the following convex program of $p \in \Delta(\mathcal{A})$ is always feasible:*

$$\forall a \in \mathcal{A} : \sum_b f(x, a, b)p(b) + \frac{2}{\gamma p(a)} \leq \frac{5A}{\gamma}.$$

*Furthermore, any solution $p$ satisfies:*

$$\mathbb{E}_{a \sim p} \Big[ f^\star(x, \pi_{f^\star}(x), a) \Big] \leq \frac{\gamma}{4} \mathbb{E}_{a,b \sim p} \Big[ \big(f(x, a, b) - f^\star(x, a, b)\big)^2 \Big] + \frac{5A}{\gamma}$$

*whenever $\gamma \geq 2A$.*

**Lemma 12.** *Assume that for each $f \in \mathcal{F}$, there exists an associated function $r : \mathcal{X} \times \mathcal{A} \to [0, 1]$ such that $f(x, a, b) = r(x, a) - r(x, b)$ for any $x \in \mathcal{X}$ and $a, b \in \mathcal{A}$. In this case, for any context $x \in \mathcal{X}$, if we define $p$ as*

$$p(a) = \begin{cases} \frac{1}{A + \gamma\big(r(x, \pi_f(x)) - r(x, a)\big)} & a \neq \pi_f(x) \\ 1 - \sum_{a \neq \pi_f(x)} p(a) & a = \pi_f(x) \end{cases},$$

*then we have*

$$\mathbb{E}_{a\sim p}\left[f^\star(x,\pi_{f^\star}(x),a)\right] \le \gamma \mathbb{E}_{a,b\sim p}\left[\left(f(x,a,b)-f^\star(x,a,b)\right)^2\right] + \frac{A}{\gamma}$$

*Proof of lemma 12.* Fix any $b\in\mathcal{A}$. Then, the distribution $p$ can be rewritten as

$$p(a) = \begin{cases} \left(A + 2\gamma\left(\frac{r(x,\pi_f(x))-r(x,b)+1}{2} - \frac{r(x,a)-r(x,b)+1}{2}\right)\right)^{-1} & a \ne \pi_f(x) \\ 1 - \sum_{a\ne\pi_f(x)} p(a) & a = \pi_f(x) \end{cases}.$$

Therefore, denoting $f^\star(x,a,b) = r^\star(x,a) - r^\star(x,b)$ for some function $r^\star$, we have

$$\mathbb{E}_{a\sim p}\left[f^\star(x,\pi_{f^\star}(x),a)\right] = \mathbb{E}_{a\sim p}\left[r^\star(x,\pi_{f^\star}(x)) - r^\star(x,a)\right]$$

$$= 2\,\mathbb{E}_{a\sim p}\left[\frac{r^\star(x,\pi_{f^\star}(x))-r^\star(x,b)+1}{2} - \frac{r^\star(x,a)-r^\star(x,b)+1}{2}\right]$$

$$\le 2\cdot 2\gamma\,\mathbb{E}_{a\sim p}\left[\left(\frac{r(x,a)-r(x,b)+1}{2} - \frac{r^\star(x,a)-r^\star(x,b)+1}{2}\right)^2\right] + \frac{A}{\gamma}$$

$$= \gamma\,\mathbb{E}_{a\sim p}\left[\left(f(x,a,b)-f^\star(x,a,b)\right)^2\right] + \frac{A}{\gamma}$$

where for the inequality above we invoked Lemma 3 with $\hat{y}(a) = (r(x,a)-r(x,b)+1)/2$ and $y^\star(a) = (r^\star(x,a)-r^\star(x,b)+1)/2$. We note that the above holds for any $b\in\mathcal{A}$. Hence, we complete the proof by sampling $b\sim p$. $\qquad\square$

**Lemma 13.** *Assume $f^\star\in\mathcal{F}_t$ for all $t\in[T]$. Suppose there exists some $t'\in[T]$ such that $\lambda_t = 0$ for all $t\le t'$. Then we have*

$$\sum_{t=1}^{t'} Z_t w_t \le 56A^2\beta\cdot\frac{\dim_E(\mathcal{F},\Delta)}{\Delta}\cdot\log(2/(\delta\Delta))$$

*with probability at least $1-\delta$.*

*Proof.* Since $f^\star\in\mathcal{F}_t$, we always have $\pi_{f^\star}(x_t)\in\mathcal{A}_t$ for all $t\in[T]$. Hence, whenever $Z_t$ is zero, we have $\mathcal{A}_t = \{\pi_{f^\star}(x_t)\}$ and thus we do not incur any regret. Hence, we know $Z_t w_t$ is either 0 or at least $\Delta$ by Lemma 7. Let us fix an integer $m > 1/\Delta$, whose value will be specified later. We divide the interval $[\Delta, 1]$ into bins of width $1/m$ and conduct a refined study of the sum of $Z_t w_t$:

$$\sum_{t=1}^{t'} Z_t w_t \le \sum_{t=1}^{t'}\sum_{j=0}^{(1-\Delta)m-1} Z_t w_t\cdot\mathbb{1}\left\{Z_t w_t\in\left[\Delta+\frac{j}{m},\,\Delta+\frac{j+1}{m}\right]\right\}$$

$$\le \sum_{j=0}^{(1-\Delta)m-1}\left(\Delta+\frac{j+1}{m}\right)\sum_{t=1}^{t'} Z_t\mathbb{1}\left\{w_t\ge\Delta+\frac{j}{m}\right\}$$

$$= \sum_{j=0}^{(1-\Delta)m-1}\left(\Delta+\frac{j+1}{m}\right)\sum_{t=1}^{t'} Z_t\mathbb{1}\left\{\sup_{a,b\in\mathcal{A}_t}\sup_{f,f'\in\mathcal{F}_t} f(x_t,a,b)-f'(x_t,a,b)\ge\Delta+\frac{j}{m}\right\}$$

$$= \sum_{j=0}^{(1-\Delta)m-1}\left(\Delta+\frac{j+1}{m}\right)\sum_{t=1}^{t'} Z_t\sup_{a,b\in\mathcal{A}_t}\mathbb{1}\left\{\sup_{f,f'\in\mathcal{F}_t} f(x_t,a,b)-f'(x_t,a,b)\ge\Delta+\frac{j}{m}\right\}$$

$$\le \sum_{j=0}^{(1-\Delta)m-1}\left(\Delta+\frac{j+1}{m}\right)\sum_{t=1}^{t'} Z_t\sum_{a,b}\mathbb{1}\left\{\sup_{f,f'\in\mathcal{F}_t} f(x_t,a,b)-f'(x_t,a,b)\ge\left(\Delta+\frac{j}{m}\right)\right\}$$

$$\le \sum_{j=0}^{(1-\Delta)m-1}\left(\Delta+\frac{j+1}{m}\right)A^2\underbrace{\sum_{t=1}^{t'} Z_t\,\mathbb{E}_{a,b\sim p_t}\mathbb{1}\left\{\sup_{f,f'\in\mathcal{F}_t} f(x_t,a,b)-f'(x_t,a,b)\ge\left(\Delta+\frac{j}{m}\right)\right\}}_{(*)}$$

where in the third inequality we replace the supremum over $a, b$ by the summation over $a, b$, and in the last inequality we further replace it by the expectation. Here recall that $p_t(a)$ is uniform when $\lambda_t = 0$, leading to the extra $A^2$ factor. To deal with $(*)$, we first apply Lemma 4 to recover the empirical $a_t$ and $b_t$, and then apply Lemma 10 to get an upper bound via the eluder dimension:

$$(*) \leq 2 \sum_{t=1}^{t'} Z_t \mathbb{1} \left\{ \sup_{f, f' \in \mathcal{F}_t} f(x_t, a_t, b_t) - f'(x_t, a_t, b_t) \geq \left( \Delta + \frac{j}{m} \right) \right\} + 8 \log(\delta^{-1})$$

$$\leq 2 \left( \frac{4\beta}{\left( \Delta + \frac{j}{m} \right)^2} + 1 \right) \dim_E \left( \mathcal{F}; \Delta \right) + 8 \log(\delta^{-1})$$

$$\leq \frac{10\beta}{\left( \Delta + \frac{j}{m} \right)^2} \cdot \dim_E \left( \mathcal{F}; \Delta \right) + 8 \log(\delta^{-1})$$

with probability at least $1 - \delta$. Plugging $(*)$ back, we obtain

$$\sum_{t=1}^{t'} Z_t w_t \leq \sum_{j=0}^{(1-\Delta)m-1} \left( \Delta + \frac{j+1}{m} \right) \cdot \frac{10A^2\beta}{\left( \Delta + \frac{j}{m} \right)^2} \cdot \dim_E \left( \mathcal{F}; \Delta \right) + 8mA^2 \log(\delta^{-1})$$

$$= 10A^2\beta \cdot \dim_E \left( \mathcal{F}, \Delta \right) \sum_{j=0}^{(1-\Delta)m-1} \frac{\Delta + \frac{j+1}{m}}{\left( \Delta + \frac{j}{m} \right)^2} + 8mA^2 \log(\delta^{-1})$$

$$\leq 10A^2\beta \cdot \dim_E \left( \mathcal{F}, \Delta \right) \left( \frac{\Delta + 1/m}{\Delta^2} + \sum_{j=1}^{(1-\Delta)m-1} \frac{2}{\Delta + \frac{j}{m}} \right) + 8mA^2 \log(\delta^{-1})$$

$$\leq 10A^2\beta \cdot \dim_E \left( \mathcal{F}, \Delta \right) \sum_{j=0}^{(1-\Delta)m-1} \frac{2}{\Delta + \frac{j}{m}} + 8mA^2 \log(\delta^{-1})$$

$$\leq 20A^2\beta \cdot \dim_E \left( \mathcal{F}, \Delta \right) \sum_{j=0}^{(1-\Delta)m-1} \int_{j-1}^{j} \frac{1}{\Delta + \frac{x}{m}} \, \mathrm{d}x + 8mA^2 \log(\delta^{-1})$$

$$= 20A^2\beta \cdot \dim_E \left( \mathcal{F}, \Delta \right) \int_{-1}^{(1-\Delta)m-1} \frac{1}{\Delta + \frac{x}{m}} \, \mathrm{d}x + 8mA^2 \log(\delta^{-1})$$

$$= 20A^2\beta \cdot \dim_E \left( \mathcal{F}, \Delta \right) \cdot m \log \left( \frac{1}{\Delta - m^{-1}} \right) + 8mA^2 \log(\delta^{-1})$$

where for the second inequality, we use the fact that $(j+1)/m \leq 2j/m$ for any $j \geq 1$; for the third inequality, we assume $m > 1/\Delta$. Setting $m = 2/\Delta$, we arrive at

$$\sum_{t=1}^{t'} Z_t w_t \leq 40A^2\beta \cdot \frac{\dim_E \left( \mathcal{F}, \Delta \right)}{\Delta} \cdot \log(2/\Delta) + 16A^2 \log(\delta^{-1})/\Delta$$

$$\leq 56A^2\beta \cdot \frac{\dim_E \left( \mathcal{F}, \Delta \right)}{\Delta} \cdot \log(2/(\delta\Delta)),$$

which completes the proof. $\qquad\square$

**Lemma 14.** *Whenever*

$$56A^2\beta \cdot \dim_E \left( \mathcal{F}, \Delta \right) \cdot \log(2/(\delta\Delta))/\Delta < \sqrt{AT/\beta},$$

*we have $\lambda_1 = \lambda_2 = \cdots = \lambda_T = 0$ with probability at least $1 - \delta$.*

*Proof of Lemma 14.* We prove it via contradiction. Assume the inequality holds but there exists $t'$ for which $\lambda_{t'} = 1$. Without loss of generality, we assume that $\lambda_t = 0$ for all $t < t'$, namely that $t'$ is the first time that $\lambda_t$ is 1. Then by definition of $\lambda_{t'}$, we have

$$\sum_{s=1}^{t'-1} Z_s w_s \geq \sqrt{AT/\beta}.$$

On the other hand, by Lemma 13, we have

$$\sum_{s=1}^{t'-1} Z_s w_s \leq 56A^2\beta \cdot \frac{\dim_E(\mathcal{F}, \Delta)}{\Delta} \cdot \log(2/(\delta\Delta)).$$

The combination of the above two inequalities contradicts with the conditions. $\square$

## C.2 Proof of Lemma 1

*Proof of Lemma 1.* We prove it via contradiction. If no such arm exists, meaning that for any arm $a$, there exists an arm $b$ such that $f^\star(x, a, b) < 0$. Then we can find a sequence of arms $(a_1, a_2, \ldots, a_k)$ such that $f^\star(x, a_i, a_{i+1}) < 0$ for any $i = 1, \ldots, k-1$ and $f^\star(x, a_k, a_1) < 0$, which contradicts with the transitivity (Assumption 1). $\square$

## C.3 Proof of Theorem 1

We begin by showing the worst-case regret upper bound.

**Lemma 15** (Worst-case regret upper bound)**.** *For Algorithm 1, assume $f^\star \in \mathcal{F}_t$ for all $t \in [T]$. Then, we have*

$$\text{Regret}_T^{\text{CB}} \leq 68\sqrt{AT\beta} \cdot \log(4\delta^{-1})$$

*with probability at least $1 - \delta$.*

*Proof of Lemma 15.* We recall that the regret is defined as

$$\text{Regret}_T^{\text{CB}} = \sum_{t=1}^{T} \left( f^\star(x_t, \pi_{f^\star}(x_t), a_t) + f^\star(x_t, \pi_{f^\star}(x_t), b_t) \right).$$

Since $a_t$ and $b_t$ are always drawn independently from the same distribution in Algorithm 1, we only need to consider the regret of the $a_t$ part in the following proof for brevity — multiplying the result by two would yield the overall regret.

We first observe the definition of $\lambda_t$ in Algorithm 1: the left term $\sum_{s=1}^{t-1} Z_s w_s$ in the indicator is non-decreasing in $t$ while the right term remains constant. This means that there exists a particular time step $t' \in [T]$ dividing the time horizon into two phases: $\lambda_t = 0$ for all $t \leq t'$ and $\lambda_t = 1$ for all $t > t'$. Now, we proceed to examine these two phases individually.

For all rounds before or on $t'$, we can compute the expected partial regret as

$$\sum_{t=1}^{t'} \mathbb{E}_{a \sim p_t} \left[ f^\star(x_t, \pi_{f^\star}(x_t), a) \right] = \sum_{t=1}^{t'} Z_t \mathbb{E}_{a \sim p_t} \left[ f^\star(x_t, \pi_{f^\star}(x_t), a) \right] \leq \sum_{t=1}^{t'} Z_t w_t \leq \sqrt{AT\beta}, \quad (8)$$

where the equality holds since we have $\mathcal{A}_t = \{\pi_{f^\star}(x_t)\}$ whenever $Z_t = 0$ under the condition that $f^\star \in \mathcal{F}_t$, and thus we don't incur regret in this case. The first inequality is Lemma 8, and the second inequality holds by the definition of $\lambda_t$ and the condition that $\lambda_t = 0$.

On the other hand, for all rounds after $t'$, we have

$$\sum_{t=t'+1}^{T} \mathbb{E}_{a\sim p_t} \left[ f^\star(x_t, \pi_{f^\star}(x_t), a) \right]$$

$$= \sum_{t=t'+1}^{T} Z_t \mathbb{E}_{a\sim p_t} \left[ f^\star(x_t, \pi_{f^\star}(x_t), a) \right]$$

$$\leq \sum_{t=t'+1}^{T} Z_t \left( \frac{5A}{\gamma_t} + \frac{\gamma_t}{4} \mathbb{E}_{a,b\sim p_t} \left[ \left( f^\star(x_t, a, b) - f_t(x_t, a, b) \right)^2 \right] \right)$$

$$= \sum_{t=t'+1}^{T} Z_t \left( \frac{5A}{\sqrt{AT/\beta}} + \frac{\sqrt{AT/\beta}}{4} \mathbb{E}_{a,b\sim p_t} \left[ \left( f^\star(x_t, a, b) - f_t(x_t, a, b) \right)^2 \right] \right)$$

$$\leq 5\sqrt{AT\beta} + \frac{\sqrt{AT/\beta}}{4} \sum_{t=t'+1}^{T} Z_t \mathbb{E}_{a,b\sim p_t} \left[ \left( f^\star(x_t, a, b) - f_t(x_t, a, b) \right)^2 \right]$$

$$\leq 5\sqrt{AT\beta} + \frac{\sqrt{AT/\beta}}{2} \sum_{t=t'+1}^{T} Z_t \left( f^\star(x_t, a_t, b_t) - f_t(x_t, a_t, b_t) \right)^2 + 8\sqrt{AT/\beta} \cdot \log(4\delta^{-1})$$

$$\leq 5\sqrt{AT\beta} + \frac{\sqrt{AT\beta}}{2} + 8\sqrt{AT/\beta} \cdot \log(4\delta^{-1}). \tag{9}$$

where the first inequality holds by Lemma 11 (or Lemma 12 for specific function classes), the second equality is by the definition of $\gamma_t$, the third inequality is by Lemma 4, and the fourth inequality holds by Lemma 9.

Putting the two parts, (8) and (9), together, we arrive at

$$\sum_{t=1}^{T} \mathbb{E}_{a\sim p_t} \left[ f^\star(x_t, \pi_{f^\star}(x_t), a) \right] \leq 7\sqrt{AT\beta} + 8\sqrt{AT/\beta} \cdot \log(4\delta^{-1}) \leq 15\sqrt{AT\beta} \cdot \log(4\delta^{-1}).$$

Now we apply Lemma 4 again. The following holds with probability at least $1 - \delta/2$,

$$\sum_{t=1}^{T} f^\star(x_t, \pi_{f^\star}(x_t), a_t) \leq 2\sum_{t=1}^{T} \mathbb{E}_{a\sim p_t} \left[ f^\star(x_t, \pi_{f^\star}(x_t), a) \right] + 4\log(4\delta^{-1}) \leq 34\sqrt{AT\beta} \cdot \log(4\delta^{-1}).$$

The above concludes the regret of the $a_t$ part. The regret of the $b_t$ can be shown in the same way. Adding them together, we conclude that

$$\text{Regret}_T^{\text{CB}} = \sum_{t=1}^{T} \left( f^\star(x_t, \pi_{f^\star}(x_t), a_t) + f^\star(x_t, \pi_{f^\star}(x_t), b_t) \right) \leq 68\sqrt{AT\beta} \cdot \log(4\delta^{-1}).$$

$\square$

**Lemma 16** (Instance-dependent regret upper bound). *For Algorithm 1, assume $f^\star \in \mathcal{F}_t$ for all $t \in [T]$. Then, we have*

$$\text{Regret}_T^{\text{CB}} \leq 3808 A^2\beta^2 \cdot \frac{\dim_E(\mathcal{F}, \Delta)}{\Delta} \cdot \log^2(4/(\delta\Delta))$$

*with probability at least $1 - \delta$.*

*Proof of Lemma 16.* We consider two cases. First, when

$$56A^2\beta \cdot \frac{\dim_E(\mathcal{F}, \Delta)}{\Delta} \cdot \log(2/(\delta\Delta)) < \sqrt{AT/\beta}, \tag{10}$$

we invoke Lemma 14 and get that $\lambda_t = 0$ for all $t \in [T]$. Hence, we have

$$
\begin{aligned}
\text{Regret}_T^{\text{CB}} &= \sum_{t=1}^{T} \left( f^\star(x_t, \pi_{f^\star}(x_t), a_t) + f^\star(x_t, \pi_{f^\star}(x_t), b_t) \right) \\
&\leq 2 \sum_{t=1}^{T} Z_t w_t \\
&\leq 112 A^2 \beta \cdot \frac{\dim_E (\mathcal{F}, \Delta)}{\Delta} \cdot \log(2/(\delta\Delta)) \\
&\leq 3808 A^2 \beta^2 \cdot \frac{\dim_E (\mathcal{F}, \Delta)}{\Delta} \cdot \log^2(4/(\delta\Delta))
\end{aligned}
$$

where the first inequality is by Lemma 8 and the fact that we incur no regret when $Z_t = 0$ since $f^\star \in \mathcal{F}_t$. The second inequality is by Lemma 13.

On the other hand, when the contrary of (10) holds, i.e.,

$$
56 A^2 \beta \cdot \frac{\dim_E (\mathcal{F}, \Delta)}{\Delta} \cdot \log(2/(\delta\Delta)) \geq \sqrt{AT/\beta}, \tag{11}
$$

applying Lemma 15, we have

$$
\begin{aligned}
\text{Regret}_T^{\text{CB}} &\leq 68 \sqrt{AT\beta} \cdot \log(4\delta^{-1}) \\
&= 68\beta \cdot \log(4\delta^{-1}) \cdot \sqrt{AT/\beta} \\
&\leq 68\beta \cdot \log(4\delta^{-1}) \cdot 56 A^2 \beta \cdot \frac{\dim_E (\mathcal{F}, \Delta)}{\Delta} \cdot \log(2/(\delta\Delta)) \\
&\leq 3808 A^2 \beta^2 \cdot \frac{\dim_E (\mathcal{F}, \Delta)}{\Delta} \cdot \log^2(4/(\delta\Delta))
\end{aligned}
$$

where we apply the condition (11) in the second inequality. $\qquad\square$

**Lemma 17** (Query complexity). *For Algorithm 1, assume $f^\star \in \mathcal{F}_t$ for all $t \in [T]$. Then, we have*

$$
\text{Queries}_T^{\text{CB}} \leq \min \left\{ T, \, 3136 A^3 \beta^3 \frac{\dim_E^2 (\mathcal{F}, \Delta)}{\Delta^2} \cdot \log^2(2/(\delta\Delta)) \right\}
$$

*with probability at least $1 - \delta$.*

*Proof of Lemma 17.* We consider two cases. First, when

$$
56 A^2 \beta \cdot \frac{\dim_E (\mathcal{F}, \Delta)}{\Delta} \cdot \log(2/(\delta\Delta)) < \sqrt{AT/\beta} \tag{12}
$$

we can invoke Lemma 14 and get that $\lambda_t = 0$ for all $t \in [T]$. Hence,

$$
\begin{aligned}
\text{Queries}_T^{\text{CB}} &= \sum_{t=1}^{T} Z_t \\
&= \sum_{t=1}^{T} Z_t \mathbb{1}\{w_t \geq \Delta\} \\
&= \sum_{t=1}^{T} Z_t \sup_{a,b \in \mathcal{A}_t} \mathbb{1} \left\{ \sup_{f,f' \in \mathcal{F}_t} f(x_t, a, b) - f'(x_t, a, b) \geq \Delta \right\} \\
&\leq \sum_{t=1}^{T} Z_t \sum_{a,b} \mathbb{1} \left\{ \sup_{f,f' \in \mathcal{F}_t} f(x_t, a, b) - f'(x_t, a, b) \geq \Delta \right\} \\
&\leq A^2 \underbrace{\sum_{t=1}^{T} Z_t \mathop{\mathbb{E}}_{a,b \sim p_t} \mathbb{1} \left\{ \sup_{f,f' \in \mathcal{F}_t} f(x_t, a, b) - f'(x_t, a, b) \geq \Delta \right\}}_{(*)}
\end{aligned}
$$

where the second equality is by Lemma 7, the second inequality holds as $p_t(a)$ is uniform for any $a, b$ when $\lambda_t = 0$. We apply Lemma 4 and Lemma 10 to $(*)$ and obtain

$$
(*) \leq 2 \sum_{t=1}^{T} Z_t \mathbb{1} \left\{ \sup_{f, f' \in \mathcal{F}_t} f(x_t, a_t, b_t) - f'(x_t, a_t, b_t) \geq \Delta \right\} + 8 \log(\delta^{-1})
$$

$$
\leq 2 \left( \frac{4\beta}{\Delta^2} + 1 \right) \dim_E(\mathcal{F}; \Delta) + 8 \log(\delta^{-1})
$$

$$
\leq \frac{10\beta}{\Delta^2} \cdot \dim_E(\mathcal{F}; \Delta) + 8 \log(\delta^{-1}).
$$

Plugging this back, we obtain

$$
\mathrm{Queries}_T^{\mathrm{CB}} \leq \frac{10 A^2 \beta}{\Delta^2} \cdot \dim_E(\mathcal{F}; \Delta) + 8 A^2 \log(\delta^{-1})
$$

$$
\leq 3136 A^3 \beta^3 \frac{\dim_E^2(\mathcal{F}, \Delta)}{\Delta^2} \cdot \log^2(2/(\delta\Delta)).
$$

On the other hand, when the contrary of (12) holds, i.e.,

$$
56 A^2 \beta \cdot \frac{\dim_E(\mathcal{F}, \Delta)}{\Delta} \cdot \log(2/(\delta\Delta)) \geq \sqrt{AT/\beta}.
$$

Squaring both sides, we obtain

$$
3136 A^4 \beta^2 \frac{\dim_E^2(\mathcal{F}, \Delta)}{\Delta^2} \cdot \log^2(2/(\delta\Delta)) \geq AT/\beta
$$

which leads to

$$
T \leq 3136 A^3 \beta^3 \frac{\dim_E^2(\mathcal{F}, \Delta)}{\Delta^2} \cdot \log^2(2/(\delta\Delta)).
$$

We note that we always have $\mathrm{Queries}_T^{\mathrm{CB}} \leq T$, and thus,

$$
\mathrm{Queries}_T^{\mathrm{CB}} \leq T \leq 3136 A^3 \beta^3 \frac{\dim_E^2(\mathcal{F}, \Delta)}{\Delta^2} \cdot \log^2(2/(\delta\Delta)).
$$

Hence, we complete the proof. $\qquad \square$

Having established the aforementioned lemmas, we are now able to advance towards the proof of Theorem 1.

*Proof of Theorem 1.* By Lemma 9 and the construction of version spaces $\mathcal{F}_t$ in Algorithm 1, we have $f^\star \in \mathcal{F}_t$ for all $t \in [T]$ with probability at least $1 - \delta$. Then, the rest of the proof follows from Lemmas 15 to 17. $\qquad \square$

## C.4 Proof of Theorem 2

In this section, we will prove the following theorem, which is stronger than Theorem 2.

**Theorem 5** (Lower bounds). *The following two claims hold:*

*(1) for any algorithm, there exists an instance that leads to* $\mathrm{Regret}_T^{\mathrm{CB}} = \Omega(\sqrt{AT})$;

*(2) for any algorithm achieving a worse-case expected regret upper bound in the form of* $\mathbb{E}[\mathrm{Regret}_T^{\mathrm{CB}}] = O(\sqrt{A} \cdot T^{1-\beta})$ *for some* $\beta > 0$*, there exists an instance with gap* $\Delta = \sqrt{A} \cdot T^{-\beta}$ *that results in* $\mathbb{E}[\mathrm{Regret}_T^{\mathrm{CB}}] = \Omega(A/\Delta) = \Omega(\sqrt{A} \cdot T^{\beta})$ *and* $\mathbb{E}[\mathrm{Queries}_T^{\mathrm{CB}}] = \Omega(A/\Delta^2) = \Omega(T^{2\beta})$.

We observe that Theorem 2 can be considered as a corollary of the above theorem when setting $\beta = 1/2$.

In what follows, we will first demonstrate lower bounds in the setting of *multi-armed bandits (MAB) with active queries* and subsequently establish a reduction from it to contextual dueling bandits in

order to achieve these lower bounds. We start by formally defining the setting of MAB with active queries below.

**Multi-armed bandits with active queries.** We consider a scenario where there exist $A$ arms. Each arm $a$ is assumed to yield a binary reward (0 or 1), which is sampled from a Bernoulli distribution $\mathrm{Bern}(\bar{r}_a)$, where $\bar{r}_a$ denotes the mean reward associated with arm $a$. The arm with the highest mean reward is denoted by $a^\star := \arg\max_a \bar{r}_a$. Let $\Delta_a := \bar{r}_{a^\star} - \bar{r}_a$ denote the gap of arm $a \in [A]$. The interaction proceeds as follows: at each round $t \in [T]$, we need to pull an arm but can choose whether to receive the reward signal (denote this choice by $Z_t$). The objective is to minimize two quantities: the regret and the number of queries,

$$\mathrm{Regret}_T = \sum_{t=1}^{T} \Delta_{a_t}, \quad \mathrm{Queries}_T = \sum_{t=1}^{T} Z_t. \tag{13}$$

Towards the lower bounds, we will start with a bound on the KL divergence over distributions of runs under two different bandits. This result is a variant of standard results which can be found in many bandit literature (e.g., Lattimore and Szepesvári [2020]).

**Lemma 18.** *Let $I_1$ and $I_2$ be two instances of MAB. We define $p_1$ and $p_2$ as their respective distributions over the outcomes of all pulled arms and reward signals when a query is made. Concretely, $p_1$ and $p_2$ are measuring the probability of outcomes (denoted by $O$) in the following form:*

$$O = \big(Z_1, a_1, (r_1), \ldots, Z_T, a_T, (r_T)\big)$$

*where the reward $r_t$ is included only when $Z_t = 1$, and we added parentheses above to indicate this point. We denote $\mathrm{Pr}_1$ (resp. $\mathrm{Pr}_2$) as the reward distribution of $I_1$ (resp. $I_2$). We define $\bar{n}_a = \sum_{t=1}^{T} Z_t \mathbb{1}\{a_t = a\}$ as the number of times arm $a$ is pulled when making a query. Then, given any algorithm $\mathfrak{A}$, the Kullback–Leibler divergence between $p_1$ and $p_2$ can be decomposed in the following way*

$$\mathrm{KL}(p_1, p_2) = \sum_{a=1}^{A} \mathbb{E}_{p_1}[\bar{n}_a] \cdot \mathrm{KL}\big(\mathrm{Pr}_1(r \,|\, a), \mathrm{Pr}_2(r \,|\, a)\big).$$

*Proof of Lemma 18.* We define the conditional distribution

$$\overline{\mathrm{Pr}}_1(r_t \,|\, Z_t, a_t) \begin{cases} \mathrm{Pr}_1(r_t \,|\, a_t) & \text{if } Z_t = 1 \\ 1 & \text{if } Z_t = 0 \end{cases},$$

and similarly for $\overline{\mathrm{Pr}}_2$. Additionally, we denote $\mathrm{Pr}_{\mathfrak{A}}$ as the probability associated with algorithm $\mathfrak{A}$. Then, for any outcome $O$, we have

$$p_1(O) = \prod_{t=1}^{T} \mathrm{Pr}_{\mathfrak{A}}\big(Z_t, a_t \,|\, Z_1, a_1, (r_1), \ldots, Z_{t-1}, a_{t-1}, (r_{t-1})\big) \overline{\mathrm{Pr}}_1(r_t \,|\, Z_t, a_t),$$

and we can write $p_2(O)$ in a similar manner. Hence,

$$
\begin{aligned}
\mathrm{KL}(p_1, p_2) &= \mathbb{E}_{O \sim p_1} \left[ \log\left( \frac{\prod_{t=1}^{T} \mathrm{Pr}_{\mathfrak{A}}\big(Z_t, a_t \,|\, Z_1, a_1, (r_1), \ldots, Z_{t-1}, a_{t-1}, (r_{t-1})\big) \overline{\mathrm{Pr}}_1(r_t \,|\, Z_t, a_t)}{\prod_{t=1}^{T} \mathrm{Pr}_{\mathfrak{A}}\big(Z_t, a_t \,|\, Z_1, a_1, (r_1), \ldots, Z_{t-1}, a_{t-1}, (r_{t-1})\big) \overline{\mathrm{Pr}}_2(r_t \,|\, Z_t, a_t)} \right) \right] \\
&= \mathbb{E}_{O \sim p_1} \left[ \sum_{t=1}^{T} \log\left( \frac{\overline{\mathrm{Pr}}_1(r_t \,|\, Z_t, a_t)}{\overline{\mathrm{Pr}}_2(r_t \,|\, Z_t, a_t)} \right) \right] \\
&= \mathbb{E}_{O \sim p_1} \left[ \sum_{t=1}^{T} Z_t \log\left( \frac{\mathrm{Pr}_1(r_t \,|\, a_t)}{\mathrm{Pr}_2(r_t \,|\, a_t)} \right) \right] \\
&= \mathbb{E}_{O \sim p_1} \left[ \sum_{t=1}^{T} Z_t \mathbb{E}_{r_t \sim \mathrm{Pr}_1(\cdot \,|\, a_t)} \left[ \log\left( \frac{\mathrm{Pr}_1(r_t \,|\, a_t)}{\mathrm{Pr}_2(r_t \,|\, a_t)} \right) \right] \right] \\
&= \mathbb{E}_{O \sim p_1} \left[ \sum_{t=1}^{T} Z_t \cdot \mathrm{KL}\big(\mathrm{Pr}_1(\cdot \,|\, a_t), \mathrm{Pr}_2(\cdot \,|\, a_t)\big) \right] \\
&= \sum_{a=1}^{A} \mathbb{E}_{O \sim p_1}[\bar{n}_a] \cdot \mathrm{KL}\big(\mathrm{Pr}_1(\cdot \,|\, a_t), \mathrm{Pr}_2(\cdot \,|\, a_t)\big)
\end{aligned}
$$

where the third equality holds by the definition of $\overline{\mathrm{Pr}}_1$ and $\overline{\mathrm{Pr}}_2$. $\qquad\square$

The following lemma establishes lower bounds for MAB with active queries. It presents a trade-off between the regret and the number of queries.

**Lemma 19.** *Let $\mathcal{I}$ denote the set of all MAB instances. Assume* ALG *is an algorithm that achieves the following worst-case regret upper bound for some $C$ and $\beta$:*

$$\mathbb{E}\left[\mathrm{Regret}_T\right] \leq CT^{1-\beta},$$

*for all $I \in \mathcal{I}$. Then, for any MAB instance $I \in \mathcal{I}$, the regret and the number of queries made by algorithm* ALG *are lower bounded:*

$$\mathbb{E}\left[\mathrm{Regret}_T\right] \geq \sum_{a \neq a^\star} \frac{\zeta}{\Delta_a} \log\left(\frac{\Delta_a}{4CT^{-\beta}}\right), \quad \mathbb{E}\left[\mathrm{Queries}_T\right] \geq \sum_{a \neq a^\star} \frac{\zeta}{\Delta_a^2} \log\left(\frac{\Delta_a}{4CT^{-\beta}}\right)$$

*where the coefficient $\zeta = \min_a \min\{\bar{r}_a, 1 - \bar{r}_a\}$ depends on the instance $I$.*

*Proof of Lemma 19.* For any MAB instance $I$ and any arm $a^\dagger$, we define a corresponding MAB instance $I'$ as follows. Denote $\bar{r}$ and $\bar{r}'$ as the mean reward of $I$ and $I'$, respectively. For $I'$, we set the mean reward $\bar{r}'_a = \bar{r}_a$ for any $a \neq a^\dagger$ and $\bar{r}'_{a^\dagger} = \bar{r}_{a^\dagger} + 2\Delta_{a^\dagger}$. Consequently, the optimal arm of $I'$ is $a^\dagger$ with margin $\Delta_{a^\dagger}$. Let $n_a$ denote the number of times that arm $a$ is pulled. We define the event

$$E = \{n_{a^\dagger} > T/2\}.$$

Then, we have

$$\mathbb{E}_p\left[\mathrm{Regret}_T\right] \geq \frac{T\Delta_{a^\dagger}}{2} \cdot p(E), \quad \mathbb{E}_{p'}\left[\mathrm{Regret}_T\right] \geq \frac{T\Delta_{a^\dagger}}{2} \cdot p'(E^{\complement}).$$

Hence,

$$\begin{aligned}
2CT^{1-\beta} &\geq \mathbb{E}_p\left[\mathrm{Regret}_T\right] + \mathbb{E}_{p'}\left[\mathrm{Regret}_T\right] \\
&\geq \frac{T\Delta_{a^\dagger}}{2}\left(p(E) + p'(E^{\complement})\right) \\
&= \frac{T\Delta_{a^\dagger}}{2}\left(1 - \left(p'(E) - p(E)\right)\right) \\
&\geq \frac{T\Delta_{a^\dagger}}{2}\left(1 - \mathrm{TV}(p, p')\right) \\
&\geq \frac{T\Delta_{a^\dagger}}{2}\left(1 - \sqrt{1 - \exp\left(-\mathrm{KL}(p, p')\right)}\right) \\
&\geq \frac{T\Delta_{a^\dagger}}{2}\exp\left(-\frac{1}{2} \cdot \mathrm{KL}(p, p')\right).
\end{aligned}$$

By Lemma 18, we have

$$\begin{aligned}
\mathrm{KL}(p, p') &= \sum_{a=1}^{A} \mathbb{E}_p[\bar{n}_a] \cdot \mathrm{KL}\left(\mathrm{Pr}(r \,|\, a), \mathrm{Pr}'(r \,|\, a)\right) \\
&= \mathbb{E}_p[\bar{n}_{a^\dagger}] \cdot \mathrm{KL}\left(\mathrm{Pr}(r \,|\, a^\dagger), \mathrm{Pr}'(r \,|\, a^\dagger)\right) \\
&\leq \mathbb{E}_p[\bar{n}_{a^\dagger}] \cdot \Delta_{a^\dagger}^2 \cdot 2/\zeta
\end{aligned}$$

where the last inequality is by Lemma 6. Putting the above two inequality together, we arrive at

$$\mathbb{E}_p[\bar{n}_{a^\dagger}] \geq \frac{\zeta}{\Delta_{a^\dagger}^2} \log\left(\frac{\Delta_{a^\dagger}}{4CT^{-\beta}}\right).$$

This establishes a query lower bound for arm $a^\dagger$. Consequently, we have

$$\mathbb{E}[\mathrm{Regret}_T] \geq \sum_{a \neq a^\star} \mathbb{E}_p[\bar{n}_a] \cdot \Delta_a \geq \sum_{a \neq a^\star} \frac{\zeta}{\Delta_a} \log\left(\frac{\Delta_a}{4CT^{-\beta}}\right),$$

and similarly,

$$\mathbb{E}[\text{Queries}_T] \geq \sum_{a \neq a^\star} \mathbb{E}[\bar{n}_a] \geq \sum_{a \neq a^\star} \frac{\zeta}{\Delta_a^2} \log\left(\frac{\Delta_a}{4CT^{-\beta}}\right).$$

$\square$

Now we can proceed with the proof of Theorem 5.

*Proof of Theorem 5.* We provide a reduction from the multi-armed bandits with active queries to the contextual dueling bandits. Our desired lower bound for the contextual dueling bandit setting thus follows from the above lower bound for Multi-Armed Bandits (MABs). Let ALG denote any algorithm for contextual dueling bandits.

**Reduction.** Since we focus on the multi-armed bandit where no context is involved, we just ignore the notation of context everywhere for brevity. We will start from an MAB instance, and then simulate a binary feedback and feed it to a dueling bandit algorithm ALG which is used to solve the original MAB instance. Particularly, consider the MAB instance with A-many actions each with an expected reward denoted as $\bar{r}_a$.

At the beginning of iteration $t$ in the MAB instance, the learner calls the dueling algorithm ALG to generate two actions $a_t$ and $b_t$. The learner plays $a_t$ at iteration $t$ to receive a reward $y_{a_t}$; the learner then moves to iteration $t + 1$ to play $b_t$, and receives reward $y_{b_t}$. At the end of iteration $t + 1$, the learner simulates a binary feedback by setting $o = 1$ if $y_{a_t} > y_{b_t}$; $o = -1$ if $y_{a_t} < y_{b_t}$; $o$ being 1 or $-1$ uniform randomly if $y_{a_t} = y_{b_t}$. Then, the learner sends $(a_t, b_t, o)$ to the dueling algorithm ALG to query for two actions which will be played at iterations $t + 2$ and $t + 3$, respectively.

From the dueling algorithm ALG's perspective, given two actions $a$ and $b$, we can verify that the probability of seeing label 1 is $(\bar{r}_a - \bar{r}_b + 1)/2$. So we can just specify the link function to be $\phi(d) = (d + 1)/2$. As we verified earlier, the corresponding $\Phi$ is strongly convex (Example 2). Moreover, since $f^\star(a, b) = \bar{r}_a - \bar{r}_b$, if we define the gap of the MAB instance as $\bar{\Delta} := \min_{a \neq a^\star}(\bar{r}_{a^\star} - \bar{r}_a)$ where $a^\star := \arg\max_i \bar{r}_i$, then we have $\bar{\Delta} = \Delta$ in this reduction where $\Delta$ is the definition of the gap in the dueling setting. We further note that the regret of the MAB instance is

$$\sum_{t=1}^{T}(\bar{r}_{a^\star} - \bar{r}_{a_t}) + \sum_{t=1}^{T}(\bar{r}_{a^\star} - \bar{r}_{b_t}),$$

which, by our definition of $f^\star$, is equivalent to the preference-based regret that occurred to the dueling algorithm ALG. The number of queries is clearly equivalent as well. Thus, the regret and the query complexity of the dueling algorithm ALG can be directly translated to the regret and the query complexity of the MAB instance.

Now, we are ready to prove the two claims in our statement.

**Proof of the first claim.** We refer the reader to Lattimore and Szepesvári [2020, Theorem 15.2] for a proof of the minimax regret lower bound of $\Omega(\sqrt{AT})$ for the MAB. Through the reduction outlined above, that lower bound naturally extends to the dueling bandits setting, yielding $\text{Regret}_T^{\text{CB}} \geq \Omega(\sqrt{AT})$ (otherwise, via the above reduction, we would have achieved an approach that breaks the lower bound of MAB).

**Proof of the second claim.** We choose an arbitrary MAB for which $\zeta = \min_a \min\{\bar{r}_a, 1 - \bar{r}_a\} > 0.2$ and the gaps of all arms are equal to $\Delta$. Invoking Lemma 19, we have

$$\mathbb{E}\left[\text{Regret}_T\right] \geq \frac{0.2(A-1)}{\Delta} \log\left(\frac{\Delta}{4CT^{-\beta}}\right) \geq \Omega\left(\frac{A}{\Delta}\right),$$

$$\mathbb{E}\left[\text{Queries}_T\right] \geq \frac{0.2(A-1)}{\Delta^2} \log\left(\frac{\Delta}{4CT^{-\beta}}\right) \geq \Omega\left(\frac{A}{\Delta^2}\right).$$

We further choose $\Delta = 40CT^{-\beta}$ and $C = \sqrt{A}$, leading to

$$\mathbb{E}\left[\text{Regret}_T\right] \geq \frac{0.2(A-1)}{40\sqrt{A}} \cdot T^\beta = \Omega\left(\sqrt{A} \cdot T^\beta\right),$$

$$\mathbb{E}\left[\text{Queries}_T\right] \geq \frac{0.2(A-1)}{1600A} \cdot T^{2\beta} = \Omega\left(T^{2\beta}\right).$$

Via the reduction we have shown above, these lower bounds naturally extend to the contextual dueling bandit setting, thereby completing the proof. □

### C.4.1 Alternative Lower Bounds Conditioning on the Limit of Regret

In this section, we establish an analogue of Theorem 5 but under a different condition. We first introduce the concept of *diminishing regret*.

**Definition 4.** *We say that an algorithm guarantees a diminishing regret if for all contextual dueling bandit instances and $p > 0$, it holds that*

$$\lim_{T \to \infty} \frac{\mathbb{E}[\mathrm{Regret}_T^{\mathrm{CB}}]}{T^p} = 0.$$

The lower bounds under the assumption of diminishing regret guarantees are stated as follows.

**Theorem 6** (Lower bounds). *The following two claims hold:*

*(1) for any algorithm, there exists an instance that leads to $\mathrm{Regret}_T^{\mathrm{CB}} \geq \Omega(\sqrt{AT})$;*

*(2) for any gap $\Delta$ and any algorithm achieving diminishing regret, there exists an instance with gap $\Delta$ that results in $\mathbb{E}[\mathrm{Regret}_T^{\mathrm{CB}}] \geq \Omega(A/\Delta)$ and $\mathbb{E}[\mathrm{Queries}_T^{\mathrm{CB}}] \geq \Omega(A/\Delta^2)$ for sufficiently large $T$.*

We should highlight that the condition of diminishing regret (Theorem 6) and the worst-case regret upper bounds (Theorems 2 and 5) are not comparable in general. However, Theorem 6 is also applicable to our algorithm (Algorithm 1) since our algorithm possesses an instance-dependent regret upper bound that is clearly diminishing.

To prove Theorem 6, we first show the following lemma, which is a variant of Lemma 19.

**Lemma 20.** *Let $\mathcal{I}$ denote the set of all MAB instances. Assume* ALG *is an algorithm that achieves diminishing regret for all MAB instances in $\mathcal{I}$, i.e., for any $I \in \mathcal{I}$ and $p > 0$, it holds that*

$$\lim_{T \to \infty} \frac{\mathbb{E}[\mathrm{Regret}_T]}{T^p} = 0.$$

*Then, for any MAB instance $I \in \mathcal{I}$, the regret and the number of queries made by algorithm* ALG *are lower bounded in the following manner:*

$$\liminf_{T \to \infty} \frac{\mathbb{E}\left[\mathrm{Regret}_T\right]}{\log T} \geq \sum_{a \neq a^\star} \frac{\zeta}{\Delta_a}, \quad \liminf_{T \to \infty} \frac{\mathbb{E}\left[\mathrm{Queries}_T\right]}{\log T} \geq \sum_{a \neq a^\star} \frac{\zeta}{\Delta_a^2}$$

*where the coefficient $\zeta := \min_a \min\{\bar{r}_a, 1 - \bar{r}_a\}$ depends on the instance $I$. Recall that $\mathrm{Regret}_T$ and $\mathrm{Queries}_T$ are defined in (13).*

*Proof of Lemma 20.* The proof is similar to Lemma 19. For any MAB instance $I \in \mathcal{I}$ and any arm $a^\dagger$, we define a corresponding MAB instance $I'$ as follows. Denote $\bar{r}$ and $\bar{r}'$ as the mean reward of $I$ and $I'$, respectively. For $I'$, we set the mean reward $\bar{r}'_a = \bar{r}_a$ for any $a \neq a^\dagger$ and $\bar{r}'_{a^\dagger} = \bar{r}_{a^\dagger} + 2\Delta_{a^\dagger}$. Consequently, the optimal arm of $I'$ is $a^\dagger$ with margin $\Delta_{a^\dagger}$. Let $n_a$ denote the number of times that arm $a$ is pulled. We define the event

$$E = \{n_{a^\dagger} > T/2\}.$$

Let $p$ and $p'$ denote the probability of $I$ and $I'$, respectively. Then, we have

$$\mathbb{E}_p\left[\mathrm{Regret}_T\right] \geq \frac{T\Delta_{a^\dagger}}{2} \cdot p(E), \quad \mathbb{E}_{p'}\left[\mathrm{Regret}_T\right] \geq \frac{T\Delta_{a^\dagger}}{2} \cdot p'(E^{\complement})$$

where $E^{\complement}$ means the complement of event $E$. Hence,

$$\underset{p}{\mathbb{E}}\left[\text{Regret}_T\right] + \underset{p'}{\mathbb{E}}\left[\text{Regret}_T\right] \geq \frac{T\Delta_{a^\dagger}}{2}\left(p(E) + p'(E^{\complement})\right)$$

$$= \frac{T\Delta_{a^\dagger}}{2}\left(1 - \left(p'(E) - p(E)\right)\right)$$

$$\geq \frac{T\Delta_{a^\dagger}}{2}\left(1 - \text{TV}(p, p')\right)$$

$$\geq \frac{T\Delta_{a^\dagger}}{2}\left(1 - \sqrt{1 - \exp\left(-\text{KL}(p, p')\right)}\right)$$

$$\geq \frac{T\Delta_{a^\dagger}}{2}\exp\left(-\frac{1}{2}\cdot\text{KL}(p, p')\right).$$

Here TV denotes the total variation distance. By Lemma 18, we have

$$\text{KL}(p, p') = \sum_{a=1}^{A}\underset{p}{\mathbb{E}}[\bar{n}_a]\cdot\text{KL}\left(\Pr(r\,|\,a), \Pr'(r\,|\,a)\right)$$

$$= \underset{p}{\mathbb{E}}[\bar{n}_{a^\dagger}]\cdot\text{KL}\left(\Pr(r\,|\,a^\dagger), \Pr'(r\,|\,a^\dagger)\right)$$

$$\leq \underset{p}{\mathbb{E}}[\bar{n}_{a^\dagger}]\cdot\Delta_{a^\dagger}^2\cdot 2/\zeta$$

where the last inequality is by Lemma 6. Putting it all together, we arrive at

$$\underset{p}{\mathbb{E}}[\bar{n}_{a^\dagger}] \geq \frac{\zeta}{\Delta_{a^\dagger}^2}\log\left(\frac{T\Delta_{a^\dagger}}{2\left(\mathbb{E}_p\left[\text{Regret}_T\right] + \mathbb{E}_{p'}\left[\text{Regret}_T\right]\right)}\right).$$

Taking the limit on both sides yields

$$\liminf_{T\to\infty}\frac{\mathbb{E}_p[\bar{n}_{a^\dagger}]}{\log T} \geq \liminf_{T\to\infty}\frac{\zeta}{\Delta_{a^\dagger}^2}\cdot\frac{\log\left(\frac{T\Delta_{a^\dagger}}{2\left(\mathbb{E}_p\left[\text{Regret}_T\right] + \mathbb{E}_{p'}\left[\text{Regret}_T\right]\right)}\right)}{\log T}$$

$$= \liminf_{T\to\infty}\frac{\zeta}{\Delta_{a^\dagger}^2}\cdot\left(1 + \underbrace{\frac{\log(\Delta_{a^\dagger}/2)}{\log T}}_{(i)} - \underbrace{\frac{\log\left(\mathbb{E}_p\left[\text{Regret}_T\right] + \mathbb{E}_{p'}\left[\text{Regret}_T\right]\right)}{\log T}}_{(ii)}\right).$$

Here the limit of (i) is clearly 0. For the limit of (ii), we note that by the definition of diminishing regret, for any $C > 0$, there exists a $T'$ such that $\mathbb{E}[\text{Regret}_T]/T^p \leq C$ for any $T > T'$. This implies

$$\frac{\log\left(\mathbb{E}_p\left[\text{Regret}_T\right] + \mathbb{E}_{p'}\left[\text{Regret}_T\right]\right)}{\log T} \leq \frac{\log\left(2CT^p\right)}{\log T} = \frac{\log(2C)}{\log T} + p$$

for any $p > 0$. Therefore, the limit of (ii) is also 0. Plugging these back, we obtain

$$\liminf_{T\to\infty}\frac{\mathbb{E}_p[\bar{n}_{a^\dagger}]}{\log T} \geq \frac{\zeta}{\Delta_{a^\dagger}^2}.$$

This establishes a query lower bound for arm $a^\dagger$. Consequently, we have

$$\liminf_{T\to\infty}\frac{\mathbb{E}[\text{Regret}_T]}{\log T} \geq \liminf_{T\to\infty}\sum_{a\neq a^\star}\frac{\mathbb{E}_p[\bar{n}_a]\cdot\Delta_a}{\log T} \geq \sum_{a\neq a^\star}\frac{\zeta}{\Delta_a},$$

and similarly,

$$\liminf_{T\to\infty}\frac{\mathbb{E}[\text{Queries}_T]}{\log T} \geq \liminf_{T\to\infty}\sum_{a\neq a^\star}\frac{\mathbb{E}_p[\bar{n}_a]}{\log T} \geq \sum_{a\neq a^\star}\frac{\zeta}{\Delta_a^2}.$$

$\square$

Now, we proceed with the proof of Theorem 6.

*Proof of Theorem 6.* The proof of the first claim is the same as Theorem 5, so we will omit it here. Let us now focus on the proof of the second claim. By Lemma 20, for any algorithm achieving diminishing regret, the following is true for any MAB instance:

$$\liminf_{T \to \infty} \frac{\mathbb{E}\left[\text{Regret}_T\right]}{\log T} \geq \sum_{a \neq a^\star} \frac{\zeta}{\Delta_a}, \quad \liminf_{T \to \infty} \frac{\mathbb{E}\left[\text{Queries}_T\right]}{\log T} \geq \sum_{a \neq a^\star} \frac{\zeta}{\Delta_a^2}.$$

We choose an arbitrary MAB for which $\zeta \geq 0.2$ and the gaps of all suboptimal arms are equal to $\Delta$. Then, for this instance, we have

$$\liminf_{T \to \infty} \frac{\mathbb{E}\left[\text{Regret}_T\right]}{\log T} \geq \frac{0.2(A-1)}{\Delta}, \quad \liminf_{T \to \infty} \frac{\mathbb{E}\left[\text{Queries}_T\right]}{\log T} \geq \frac{0.2(A-1)}{\Delta^2}.$$

By the definition of limit, when $T$ is large enough (exceeding a certain threshold), we have

$$\frac{\mathbb{E}\left[\text{Regret}_T\right]}{\log T} \geq \frac{0.1(A-1)}{\Delta}, \quad \frac{\mathbb{E}\left[\text{Queries}_T\right]}{\log T} \geq \frac{0.1(A-1)}{\Delta^2}.$$

Via the reduction we have shown in the proof of Theorem 5, these lower bounds naturally extend to the contextual dueling bandit setting, thereby completing the proof. $\qquad\square$

## C.5 Proof of Theorem 3

*Proof of Theorem 3.* We establish the bounds for regret and the number of queries, consecutively. First, we set an arbitrary gap threshold $\epsilon > 0$. Since our algorithm is independent of $\epsilon$, we can later choose any $\epsilon$ that minimizes the upper bounds.

**Proof of regret.** We start with the regret upper bound. By definition, we have

$$\text{Regret}_T^{\text{CB}} = \sum_{t=1}^{T} \left( f^\star(x_t, \pi_{f^\star}(x_t), a_t) + f^\star(x_t, \pi_{f^\star}(x_t), b_t) \right).$$

Since $a_t$ and $b_t$ are always drawn independently from the same distribution in Algorithm 1, we only need to consider the regret of the $a_t$ part in the following proof for brevity — multiplying the result by two would yield the overall regret.

The worst-case regret upper bound presented in Lemma 15 doesn't reply on the gap assumption and thus remains applicable in this setting. Hence, we only need to prove the instance-dependent regret upper bound. To that end, we first need an analogue of Lemma 14.

**Lemma 21.** *Fix any $\epsilon > 0$. Whenever*

$$2T_\epsilon + 56A^2\beta \cdot \frac{\dim_E\left(\mathcal{F}, \epsilon\right)}{\epsilon} \cdot \log(2/(\delta\epsilon)) < \sqrt{AT/\beta},$$

*we have $\lambda_1 = \lambda_2 = \cdots = \lambda_T = 0$ with probability at least $1 - \delta$.*

*Proof of Lemma 21.* The proof is similar to Lemma 14 and is via contradiction. Assume the inequality holds but there exists $t'$ for which $\lambda_{t'} = 1$. Without loss of generality, we assume that $\lambda_t = 0$ for all $t < t'$, namely that $t'$ is the first time that $\lambda_t$ is 1. Then by definition of $\lambda_{t'}$, we have

$$\sum_{s=1}^{t'-1} Z_s w_s \geq \sqrt{AT/\beta}.$$

On the other hand, we have

$$\sum_{s=1}^{t'-1} Z_s w_s = \sum_{s=1}^{t'-1} \mathbb{1}\{\text{Gap}(x_t) \leq \epsilon\} Z_s w_s + \sum_{s=1}^{t'-1} \mathbb{1}\{\text{Gap}(x_t) > \epsilon\} Z_s w_s$$

$$\leq 2T_\epsilon + 56A^2\beta \cdot \frac{\dim_E\left(\mathcal{F}, \epsilon\right)}{\epsilon} \cdot \log(2/(\delta\epsilon))$$

where the inequality is by Lemma 13. The above two inequalities contradicts with the conditions. $\quad\square$

Towards an instance-dependent regret upper bound, we adapt the proof of Lemma 16 to this setting. We consider two cases. First, when

$$2T_\epsilon + 56A^2\beta \cdot \frac{\dim_E(\mathcal{F}, \epsilon)}{\epsilon} \cdot \log(2/(\delta\epsilon)) < \sqrt{AT/\beta}, \tag{14}$$

we invoke Lemma 21 and get that $\lambda_t = 0$ for all $t \in [T]$. Hence, we have

$$
\begin{aligned}
\mathrm{Regret}_T^{\mathrm{CB}} &= \sum_{t=1}^{T} \left( f^\star(x_t, \pi_{f^\star}(x_t), a_t) + f^\star(x_t, \pi_{f^\star}(x_t), b_t) \right) \\
&\leq 2\sum_{t=1}^{T} \mathbb{1}\{\mathrm{Gap}(x_t) \leq \epsilon\} Z_t w_t + 2\sum_{t=1}^{T} \mathbb{1}\{\mathrm{Gap}(x_t) > \epsilon\} Z_t w_t \\
&\leq 4T_\epsilon + 112A^2\beta \cdot \frac{\dim_E(\mathcal{F}, \epsilon)}{\epsilon} \cdot \log(2/(\delta\epsilon)) \\
&\leq 136\beta \cdot \log(4\delta^{-1}) \cdot T_\epsilon + 3808A^2\beta^2 \cdot \frac{\dim_E(\mathcal{F}, \epsilon)}{\epsilon} \cdot \log^2(4/(\delta\epsilon))
\end{aligned}
$$

where the first inequality is by Lemma 8 and the fact that we incur no regret when $Z_t = 0$ since $f^\star \in \mathcal{F}_t$. The second inequality is by Lemma 13.

On the other hand, when the contrary of (14) holds, i.e.,

$$2T_\epsilon + 56A^2\beta \cdot \frac{\dim_E(\mathcal{F}, \epsilon)}{\epsilon} \cdot \log(2/(\delta\epsilon)) \geq \sqrt{AT/\beta}, \tag{15}$$

applying Lemma 15, we have

$$
\begin{aligned}
\mathrm{Regret}_T^{\mathrm{CB}} &\leq 68\sqrt{AT\beta} \cdot \log(4\delta^{-1}) \\
&= 68\beta \cdot \log(4\delta^{-1}) \cdot \sqrt{AT/\beta} \\
&\leq 68\beta \cdot \log(4\delta^{-1}) \cdot \left( 2T_\epsilon + 56A^2\beta \cdot \frac{\dim_E(\mathcal{F}, \epsilon)}{\epsilon} \cdot \log(2/(\delta\epsilon)) \right) \\
&\leq 136\beta \cdot \log(4\delta^{-1}) \cdot T_\epsilon + 3808A^2\beta^2 \cdot \frac{\dim_E(\mathcal{F}, \epsilon)}{\epsilon} \cdot \log^2(4/(\delta\epsilon))
\end{aligned}
$$

where we apply the condition (15) in the second inequality.

**Proof of the number of queries.** To show an upper bound for the number of queries, we also consider two cases. First, when

$$2T_\epsilon + 56A^2\beta \cdot \frac{\dim_E(\mathcal{F}, \epsilon)}{\epsilon} \cdot \log(2/(\delta\epsilon)) < \sqrt{AT/\beta}, \tag{16}$$

we can invoke Lemma 21 and get that $\lambda_t = 0$ for all $t \in [T]$. Hence, similar to the proof of Lemma 17, we have

$$
\begin{aligned}
\mathrm{Queries}_T^{\mathrm{CB}} &= \sum_{t=1}^{T} Z_t \\
&= \sum_{t=1}^{T} Z_t \mathbb{1}\{\mathrm{Gap}(x_t) < \epsilon\} + \sum_{t=1}^{T} Z_t \mathbb{1}\{\mathrm{Gap}(x_t) \geq \epsilon\} \\
&= T_\epsilon + \sum_{t=1}^{T} Z_t \sup_{a,b \in \mathcal{A}_t} \mathbb{1}\left\{ \sup_{f,f' \in \mathcal{F}_t} f(x_t, a, b) - f'(x_t, a, b) \geq \epsilon \right\} \\
&\leq T_\epsilon + \sum_{t=1}^{T} Z_t \sum_{a,b} \mathbb{1}\left\{ \sup_{f,f' \in \mathcal{F}_t} f(x_t, a, b) - f'(x_t, a, b) \geq \epsilon \right\} \\
&\leq T_\epsilon + A^2 \underbrace{\sum_{t=1}^{T} Z_t \mathop{\mathbb{E}}_{a,b \sim p_t} \mathbb{1}\left\{ \sup_{f,f' \in \mathcal{F}_t} f(x_t, a, b) - f'(x_t, a, b) \geq \epsilon \right\}}_{(*)}
\end{aligned}
$$

where the second inequality holds as $p_t(a)$ is uniform for any $a, b$ when $\lambda_t = 0$. We apply Lemma 4 and Lemma 10 to $(*)$ and obtain

$$(*) \leq 2 \sum_{t=1}^{T} Z_t \mathbb{1} \left\{ \sup_{f, f' \in \mathcal{F}_t} f(x_t, a_t, b_t) - f'(x_t, a_t, b_t) \geq \epsilon \right\} + 8 \log(\delta^{-1})$$

$$\leq 2 \left( \frac{4\beta}{\epsilon^2} + 1 \right) \dim_E(\mathcal{F}; \epsilon) + 8 \log(\delta^{-1})$$

$$\leq \frac{10\beta}{\epsilon^2} \cdot \dim_E(\mathcal{F}; \epsilon) + 8 \log(\delta^{-1}).$$

Plugging this back, we obtain

$$\text{Queries}_T^{\text{CB}} \leq T_\epsilon + \frac{10 A^2 \beta}{\epsilon^2} \cdot \dim_E(\mathcal{F}; \epsilon) + 8 A^2 \log(\delta^{-1})$$

$$\leq 8 T_\epsilon^2 \beta / A + 6272 A^3 \beta^3 \frac{\dim_E^2(\mathcal{F}, \epsilon)}{\epsilon^2} \cdot \log^2(2/(\delta\epsilon))$$

where the second line corresponds to the upper bound derived from the alternative case, which is shown below.

When the contrary of (16) holds, i.e.,

$$2 T_\epsilon + 56 A^2 \beta \cdot \frac{\dim_E(\mathcal{F}, \epsilon)}{\epsilon} \cdot \log(2/(\delta\epsilon)) \geq \sqrt{AT/\beta}.$$

Squaring both sides and leveraging the inequality $(a + b)^2 \leq 2a^2 + 2b^2$, we obtain

$$8 T_\epsilon^2 + 6272 A^4 \beta^2 \frac{\dim_E^2(\mathcal{F}, \epsilon)}{\epsilon^2} \cdot \log^2(2/(\delta\epsilon)) \geq AT/\beta$$

which leads to

$$T \leq 8 T_\epsilon^2 \beta / A + 6272 A^3 \beta^3 \frac{\dim_E^2(\mathcal{F}, \epsilon)}{\epsilon^2} \cdot \log^2(2/(\delta\epsilon)).$$

We note that we always have $\text{Queries}_T^{\text{CB}} \leq T$ and thus

$$\text{Queries}_T^{\text{CB}} \leq T \leq 8 T_\epsilon^2 \beta / A + 6272 A^3 \beta^3 \frac{\dim_E^2(\mathcal{F}, \epsilon)}{\epsilon^2} \cdot \log^2(2/(\delta\epsilon)).$$

**Minimizing on $\epsilon$.** Given that the aforementioned proofs hold for any threshold $\epsilon$, we can select the specific value of $\epsilon$ that minimizes the upper bounds. Hence, we deduce the desired result. $\square$

### C.6 Proof of Theorem 4

*Proof of Theorem 4.* The upper bound of the number of queries is straightforward: Algorithm 2 is simply running $H$ instances of Algorithm 1, so the total number of queries is simply the sum of these $H$ instances. For bounding the regret, we have

$$\text{Regret}_T^{\text{IL}} = \sum_{t=1}^{T} V_0^{\pi_e}(x_{t,0}) - V_0^{\pi_t}(x_{t,0})$$

$$\leq \sum_{h=0}^{H-1} \sum_{t=1}^{T} \mathop{\mathbb{E}}_{x_{t,h}, a_{t,h} \sim d_{x_{t,0},h}^{\pi_t}} \left[ Q_h^{\pi_e}(x_{t,h}, \pi_h^{\pi_e}(x_{t,h})) - Q_h^{\pi_e}(x_{t,h}, a_{t,h}) \right]$$

$$\leq \sum_{h=0}^{H-1} \sum_{t=1}^{T} \mathop{\mathbb{E}}_{x_{t,h}, a_{t,h} \sim d_{x_{t,0},h}^{\pi_t}} \left[ Q_h^{\pi_e}(x_{t,h}, \pi_h^+(x_{t,h})) - Q_h^{\pi_e}(x_{t,h}, a_{t,h}) \right]$$

$$- \sum_{h=0}^{H-1} \sum_{t=1}^{T} \mathop{\mathbb{E}}_{x_{t,h} \sim d_{x_{t,0},h}^{\pi_t}} \left[ A_h^{\pi_e}(x_{t,h}, \pi_h^+(x_{t,h})) \right]$$

$$\leq H \cdot \mathbb{E} \left[ \text{Regret}_T^{\text{CB}} \right] - \text{Adv}_T.$$

where the first inequality holds by Lemma 5, and we denote $\pi_h^+(x_{t,h}) = \arg\max_a Q_h^{\pi_e}(x_{t,h}, a)$ in the second inequality. Then, we can plug the upper bound of $\text{Regret}_T^{\text{CB}}$ (Theorem 1). Moreover, we need to take a union bound over all $h \in [H]$. $\square$

