# OpenReview forum: "Contextual Bandits and Imitation Learning with Preference-Based Active Queries"
_NeurIPS.cc/2023/Conference — NeurIPS 2023 poster_

### Official Review · Reviewer_CTw4 · 2023-07-05

**Soundness:** 3 good
**Presentation:** 3 good
**Contribution:** 2 fair
**Rating:** 6
**Confidence:** 3

**Summary:**

This paper considers the learning problem of contextual bandits and imitation learning, where the learner lacks direct knowledge of the executed actions's reward (feedback), instead, the learner is only able to request the expert at each round to compare two actions.

[Interaction Protocol] The interaction between the learner and the environment proceeds in rounds with $T$ being the total number of interactions. In each round $t$, the learner first receives the context $x_t$ (which is drawn adversarially), decide whether to send request to the expert, and selects the actions (a pair of actions in the contextual bandit setting as shown in Algorithm 1).

[Preference, Request and General Function Class] For the request associated with a pair of actions $(a_t,b_t)$, the feedback $y_t \in \{ -1,+1}$ indicates either $a_t$ or $b_t$ is better, which follows an unknown preference function $f^\star$ (defined in Line 122). The learner has access to a general function class $\mathcal{F}$ where $f^\star \in \mathcal{F}$, as stated in Assumption 1.

[Goal] The performance of the learner is measured by (see Line 143 for the detailed definition): 1) the regret she suffered, that is, the difference between her total loss, and that of the optimal action; 2) the number of requests sent by the learner.

[Result of Contextual Bandits] With specific link function and online regression oracle defined in Assumption 2, Theorem 1 ensures that the regret is bounded by $\widetilde{\mathcal{O}}(\min\{ \sqrt{T}, \frac{d}{\Delta} \})$ where $d$ is some sort of complexity measurement (Eluder dimension) and $\Delta$ is the uniform gap (the minimal gap between the best and the second best action over all the context) defined in Assumption 3. The number of queries is also bounded as shown in Theorem 1. This result further matches the lower bound up to some factors (see Theorem 2 for more details).

[Imitation Result] Theorem 4 states the result and the details are deferred to Appendix, which I don't have much time to check carefully.

**Strengths:**

The paper is clearly-written, well-organized and rigorous.

1. The proof is self-contained. I haven't observed any mistakes in the lemmas I skimmed.
2. The notations, together with their meanings, are well explained. T
3. The definitions and assumptions are carefully separated.

**Weaknesses:**

1. Considering the dueling bandit problem, a special case of the problem instance studied , does the regret bound stated in Theorem 1 matches the optimal regret bound for dueling bandits? I am not quite certain about the scale of $dim_{E}(\mathcal{F}, \frac{\Delta}{2A^2})$ in this case.
2. The comparisons between the Theorem 1 and the results of Saha and Krishnamurthy [2022], and Foster et al. [2020] are not very detailed. What's regret bound of the naive conversion of ADACB into regret minimization problem, and how does it relate to this work?

**Questions:**

My questions is raised in the weakness section. I am willing to re-evaluate the score if the questions are answered properly.

**Limitations:**

This work is pure theoretical, and does not have any potential negative societal impact.

---

> ### Author Rebuttal · Authors · 2023-08-09
>
> Thank you for your valuable feedback! We would like to address your concerns as follows.
>
> **1. Comparison to regret bound for dueling bandits**
>
> As established by prior works [1,2], for dueling bandits, the minimax regret rate is $\tilde\Theta(\sqrt{AT})$ and the instance-dependent regret rate is $\tilde\Theta\left(\frac{A}{\Delta}\right)$. Now we reduce our result (theorem 1) into the dueling bandits setting and get
> $$
> \mathrm{Regret}\_T
> \leq \tilde{O}\left(\min\left\\{\sqrt{AT},\frac{A^2 \mathrm{dim}\_E(\mathcal{F},\frac{\Delta}{2A^2})}{\Delta}\right\\}\right)
> \leq \tilde{O}\left(\min\left\\{\sqrt{AT},\frac{A^3}{\Delta}\right\\}\right)
> $$
> where the second inequality holds since the eluder dimension is upper bounded by $A$ for dueling bandits. Consequently, we observe a gap of $A^2$ in the instance-dependent bound between our current rate and the optimal rate of dueling bandits. We believe that the improvement of this gap is an important future direction, and we will add this discussion to the next version of our paper.
>
> **2. comparisons between Theorem 1 and the results of Saha and Krishnamurthy [2022], and Foster et al. [2020] / regret bound of the naive conversion of ADACB into regret minimization problem**
>
> **Comparison to MinMaxCB [Saha and Krishnamurthy, 2022]**: In their setting, they assume that the preference-based feedback is sampled from $(f^\star(x,a,b)+1)/2$, which is a special case of our model (Example 2). Regarding their theoretical results, their regret upper bound is
> $$
> \mathrm{Regret}_T\leq O\left(\sqrt{AT\beta}\right)
> $$
> when translated into our notations. This is precisely identical to our worst-case regret upper bound (Theorem 1). However, we improve upon their results by having an additional instance-dependent regret bound. In other words, our algorithm can surpass theirs when the underlying contextual bandit problem exhibits good structure (e.g., small eluder dimension and large gap). Moreover, our algorithm is designed to actively make queries, and we established a guarantee on the number of queries made. In contrast, their algorithm simply queries every round.
>
> **Comparison to AdaCB [Foster et al., 2020]**: Our work shares some similarities with AdaCB, especially in terms of the form of theoretical results, but it differs in three key aspects:
>
> (1) They assume regular contextual bandits where the learner observes the reward directly, while we assume preference-based feedback. Notably, we have a reduction from "learning from reward signal" (their setting) to "learning from preference-based feedback" (our setting) when the reward is Bernoulli and the learner can choose two actions under the same state. To be specific, the reduction works as follows: assume a regular contextual bandit instance and a contextual dueling bandits algorithm. Each time the algorithm generates a comparison query between two actions, we sample the rewards of the two actions from the CB instance and return the action with the higher reward (and we return either action with equal probability in case of a tie). A detailed explanation of this reduction can be found in Appendix A.4. Thus, our setting can capture theirs under such conditions. As clear from the reduction, the regret upper bound of our algorithm remains unchanged when we convert our setting to theirs, i.e., the regret upper bound remains as follows:
> $$
> \mathrm{Regret}\_T \leq \tilde{O}\left(\min\left\\{\sqrt{AT},\frac{A^2\mathrm{dim}\_E}{\Delta}\right\\}\right),
> $$
> while for their algorithm, their proposed regret upper bound is
> $$
> \mathrm{Regret}\_T \leq \tilde{O}\left(\min\left\\{\sqrt{AT},\frac{A \theta^{\mathrm{val}}}{\Delta}\right\\}\right),
> $$
> where $\theta^{\mathrm{val}}$ denotes the *value function disagreement coefficient*. We note that comparing the eluder dimension and the value function disagreement coefficient is not straightforward since the disagreement coefficient is for stochastic settings while the eluder dimension is for adversarial settings. However, we may still observe a gap of $A$ in the instance-dependent bound. Improving upon this factor is an interesting future direction. Moreover, it is important to note that although a reduction from their setting to ours is already established in our work, the reverse direction (i.e., the reduction from ours to theirs) remains unclear and may require further investigation.
>
> (2) They assume a stochastic setting where contexts are drawn i.i.d., but we assume the context is adversarially chosen. This difference leads to distinct complexity measures in the regret upper bounds: ours involves the eluder dimension, while theirs involves the disagreement coefficient. We are not sure if these two quantities are directly comparable, and we believe that extending our algorithm to the stochastic setting to get a dependence on the disagreement coefficient is an interesting future direction.
>
> (3) It should also be noted that AdaCB does not aim to minimize query complexity, while we consider minimizing query complexity as an important goal.
>
> We will incorporate the above discussion into the next version of our paper.
>
> ---
>
> [1] Yue, Yisong, et al. "The k-armed dueling bandits problem." Journal of Computer and System Sciences 78.5 (2012): 1538-1556.
>
> [2] Saha, Aadirupa, and Pierre Gaillard. "Versatile Dueling Bandits: Best-of-both-World Analyses for Online Learning from Preferences." *ICML 2022-39th International Conference on Machine Learning*. 2022.

---

> > ### Comment · Reviewer_CTw4 · 2023-08-17
> > **Thank the authors for their response**
> >
> > Thank the authors for their response. I would like to keep the current score.

---

### Official Review · Reviewer_PzSz · 2023-07-05

**Soundness:** 3 good
**Presentation:** 3 good
**Contribution:** 3 good
**Rating:** 6
**Confidence:** 3

**Summary:**

This paper studies the contextual bandit and imitation learning problem with preference-based feedback. The authors propose an oracle-based contextual bandit algorithm, which attains both worst-case and instance-dependent regret bounds. Besides, the algorithm has an instance-dependent guarantee on the querying numbers of the preference-based information. Furthermore, the proposed bandit algorithm is extended to the imitation learning setting with provable guarantees.

**Strengths:**

- the proposed method has strong theoretical guarantees on the regret (both worst-case and instance-dependent bound) and query complexity. Although the oracle-based algorithm proposed shares similar techniques with MinMaxDB [Saha and Krishnamurthy, 2022] and AdaCB [Foster et al., 2020], the authors provide enough discussion to highlight the difference.
- lower bounds are provided to justify the upper bounds on regret, and query complexity is tight up to logarithmic factors
- the paper is well-structured and written

**Weaknesses:**

- about the practical implementation of the proposed method: one of my main concerns about the paper is from the practical side. Similar to the oracle-based algorithm for the standard contextual bandit problem (e.g., SquareCB [Foster et al. 2022]), the proposed method is established on an online regression solver with regret guarantees. However, I'm not sure to what extent such an online regression solver can be obtained with the preference-based feedback model. For instance, as shown in example 1, $f(x, a,b) = r(a,x)-r(x,b)$, the function $f(\cdot)$ is not convex even $r:\mathcal{X}\times\mathcal{A}\rightarrow[0,1]$ is a convex function, and the algorithm developed for online convex optimization is not applicable. I think it would be beneficial if the authors could provide some concrete examples (for example, the reward function has a linear structure?) that the online regression oracle is available.

-  about the instance-dependent bound: the proposed instance-dependent regret bound as an $O(\Upsilon^2)$ dependence on the regret of the oracle and an  $O(\Upsilon^3)$ on the query complexity. There seems still some room for improvement. In the finite function space case, AdaCB attains an $O(\log \vert\mathcal{F}\vert/\Delta)$ bound for a standard contextual bandit problem, but the result obtained in this paper implies an $O(\log^2 \vert\mathcal{F}\vert/\Delta)$ regret bound.




**Questions:**

- could you provide concrete examples of the online regression oracle for the preference-based feedback model? It would be even better if the author could provide more detailed discussions on to which extent such an online regression solver can be established.

- could you provide more discussion on the tightness of the instance-dependent bound, especially on the dependence of $\Upsilon$?

- The expert policy $\pi_e$ is not formally defined. Does $\pi_e$ refer to the policy that can maximize the value function? I am confused by the claim, "our algorithm not only competes with the expert policy but can also surpass it to some extent" in line 343. What is the formal definition of "surpass." Do you mean the regret would go negative due to the term $Adv_T$? However, it is unclear to me when the negative term is large enough to cancel the $O(\sqrt{T}, A/\Delta)$ term.

**Limitations:**

The paper has discussed the limitation and potential future work in the conclusion. Another issue is that it imposes a realizable assumption for $f^\star$. It is unclear whether extending the analysis for standard contextual bandit (Section 5 in [Foster et al., ICML 2020]) to the contextual dueling bandit setting is possible.

---

> ### Author Rebuttal · Authors · 2023-08-09
>
> Thank you for your valuable feedback! We would like to address your concerns as follows.
>
> **1.  Practical implementation of the online regression solver /  concrete examples of the online regression oracle**
>
> As a concrete example, when the reward function $r:\mathcal{X}\times\mathcal{A}\rightarrow[0,1]$ is linear, the function $f$ is also linear. In this case, when the loss function is chosen to be convex w.r.t. $f$ (such as square loss and log loss), the online regression oracle can be simply implemented by the online gradient descent. In addition, if $f$ is represented by a matrix (e.g., [4]) and cannot be decomposed into the difference in reward, standard convex optimization methods can still apply.
>
> At a high level, many loss functions, including square loss and log loss, are convex functionals with respect to $f$. Let us assume that $f$ is further parameterized by $\theta$. Even if the loss is not convex w.r.t. $\theta$, we may still apply non-convex programming algorithms such as non-convex FTPL [1].
>
> Moreover, many existing works have explored online regression in various scenarios. For instance, [2] and [3] have investigated online regression with square loss and general function classes. It would be interesting to integrate these works into our method.
>
> **2. Improvement on $\Upsilon$ for the instance-dependent bounds / Tightness of the instance-dependent bound**
>
> As implied by our lower bound results (Theorem 2 on page 7 and Theorem 5 in the appendix), the proposed algorithm has a regret upper bound that is tight in the gap $\Delta$ and $T$ up to logarithmic factors for both regret and query complexity. Nevertheless, we are not sure whether the dependence on other factors is tight.
>
> Specifically, we are also not sure about the tightness of $\Upsilon$. In the current work, our focus is mainly on establishing bounds up to polynomial factors on the oracle's regret.  However, we also emphasize that $\Upsilon$ is mild in most cases and usually scales like $O(\log T)$ or $O(\log|\mathcal{F}|)$ (see Examples 2 and 3). We believe that improving the algorithm to exhibit linear dependence on $\Upsilon$ is an interesting future research direction, and it seems to require non-trivial modification to the algorithm.
>
> **3. On the expert policy $\pi_e$**
>
> The expert policy $\pi_e$ can be any Markovian policy that maps from state to a distribution over actions and is unnecessarily maximizing the value function (i.e., the expert policy could be sub-optimal).
>
> Regarding the notion of "surpassing" the expert policy, we can illustrate it by considering the average regret of imitation learning, which is defined as
> $$
> \mathrm{Regret}^{\mathrm{IL}}\_T := \frac{1}{T} \sum\_{t=1}^T \Big(V^{\pi\_e}\_0(x\_{t,0}) - V^{\pi\_t}\_0(x\_{t,0})\Big).
> $$
> Then, the regret upper bound in Theorem 4 can be translated into:
> $$\mathrm{Regret}^{\mathrm{IL}}\_T\leq O\left(
> H\sqrt{\frac{A\beta}{T}}
> \right)
> -\frac{\mathrm{Adv}\_T}{T}$$
> where we have simplified it by ignoring the instance-dependent upper bound and logarithmic factors for clarity. Now, consider a case where $\max_a A^{\pi_e}_h(x, a) > \alpha_0>0$ for some constant $\alpha_0$ for all $x$ and $h$. This can happen when the expert policy is not optimal for every state. Consequently, we have $\mathrm{Adv}_T > \alpha_0 H T$. In this case, the aforementioned regret is further bounded by
> $$\mathrm{Regret}^{\mathrm{IL}}\_T \leq O\left(
> H\sqrt{\frac{A\beta}{T}}
> \right)
> -\alpha\_0 H$$
> We note that, when $T\rightarrow\infty$, we have $\mathrm{Regret}^{\mathrm{IL}}_T\rightarrow-\alpha_0 H < 0$. This means that the best policy learned in $T$ rounds will eventually outperform (or surpass) the expert policy when $T$ is large enough.
>
> We will add the above explanation to the next version to improve the clarity.
>
> **4. realizable assumption for $f^\star$ / extending [Foster et al., ICML 2020] to the contextual dueling bandit**
>
> It is a good point and an interesting question for future research. However, without a realizable assumption, we are not sure if our query complexity result still holds, although we expect that the worst-case regret bound will persist by suffering an additive term related to the model-misspecification error. We will mention it in the next version.
>
> ---
>
> [1] Agarwal, Naman, Alon Gonen, and Elad Hazan. "Learning in non-convex games with an optimization oracle." *Conference on Learning Theory*. PMLR, 2019.
>
> [2] Rakhlin, Alexander, and Karthik Sridharan. "Online non-parametric regression." *Conference on Learning Theory*. PMLR, 2014.
>
> [3] Rakhlin, Alexander, and Karthik Sridharan. "Online nonparametric regression with general loss functions." *arXiv preprint arXiv:1501.06598* (2015).
>
> [4] Saha, Aadirupa, and Akshay Krishnamurthy. "Efficient and optimal algorithms for contextual dueling bandits under realizability." International Conference on Algorithmic Learning Theory. PMLR, 2022.

---

> > ### Comment · Reviewer_PzSz · 2023-08-18
> >
> > Thank you for the detailed response. I would like to keep the positive score for this paper.

---

### Official Review · Reviewer_K3eU · 2023-07-07

**Soundness:** 3 good
**Presentation:** 3 good
**Contribution:** 2 fair
**Rating:** 6
**Confidence:** 3

**Summary:**

The paper gives “best-of-both-worlds” results for an imitation-learning problem in contextual bandits and MDP settings. With small orthogonal changes to assumptions, the algorithms primarily improve over prior work by considering instance-optimal bounds both in regret and queries, and require only ordinal preference feedback rather than explicit rewards (similar to the “dueling bandits“ literature).

**Strengths:**

- The paper is easy to read, the algorithms and notation are well-explained, and the results are appropriately contextualized in prior work.
- The examples given for the functions in the model are quite useful for grounding the problem in more concrete applications. Related work is discussed thoroughly.
- Conceptually, the model draws nice connections between contextual bandits and modern topics in finetuning models (e.g. LLMs) from preference feedback, where the emphasis on “instance-optimal” style results is particularly well-motivated.

**Weaknesses:**

- While the application of techniques from online reinforcement learning to obtain the instance-optimal bounds in this setting is clever, it is unclear how much of this follows directly vs what technical innovation is required. It would be helpful to highlight the methodological contributions used.
- Given the applications discussed, it would be beneficial to give experimental results for preference finetuning (even in a toy setting) to demonstrate the importance of instance-optimality in practice.
- While the instance-optimal rates seem reasonable, it would be nice to include (partially) matching lower bounds for some results, or discuss barriers to obtaining such results.

**Questions:**

- Can the rates on $d$ or $\Delta$ be shown to be asymptotically tight for either queries or regret?
- What does the notation $P_t[a_t, b_t]$ on line 146 refer to?

**Limitations:**

- Connections to prior RL work which makes use of eluder dimension could be discussed in greater detail.
- Some hyperlinks are broken in the PDF.

---

> ### Author Rebuttal · Authors · 2023-08-09
>
> Thank you for your valuable feedback! We would like to address your concerns as follows.
>
> **1. Highlight of the methodological contributions used.**
>
> We highlight some of the novelty and methodological contributions of the proposed algorithm below:
>
> - Active learning via candidate arm set: while the concept of a candidate arm set is not new and was originally employed in the "active arm elimination" algorithm, it is important to highlight that the integration of the candidate arm set with the active learning condition ($Z_t=\mathbf{1}\\{|\mathcal{A}_t|>1\\}$) is new to the best of our knowledge.
>
> - Best-of-both-worlds regret upper bound via well-designed query strategy: we carefully designed the query strategies for different situations (for $\lambda_t=0$ and $1$). Such a design leads to both worst-case upper bound and an instance-dependent upper bound that depends on the eluder dimension.
> - The design of the estimated cumulative regret: the quantity $\sum_{s=1}^{t-1} Z_s w_s$  (Line 9, Algorithm 1) is an upper bound of regret that we have incurred up to round $t$, which, to the best of our knowledge, is a novel contribution in the context of adversarial bandits. This is beyond another design proposed in [1], which is limited to stochastic bandits.
>
> **2. Experimental results for preference finetuning**
>
> We also believe that the experimental results of the proposed algorithm are important. Since this work primarily focuses on theoretical aspects, we leave the empirical study to future work.
>
> **3. Matching lower bounds / the asymptotical rates of $d$ and $\Delta$**
>
> We established some lower-bound results (see Theorem 2 on page 7 and Theorem 5 in the appendix). In summary, these results demonstrate the following two key points:
>
> (1) The worst-case regret is lower bounded by $\Omega(\sqrt{AT})$.
>
> (2) Any algorithm achieving a regret upper bound of $O(\sqrt{AT})$ will inevitably have an instance-dependent regret lower bounded by $\Omega(A/\Delta)$ and a query complexity lower bounded by $\Omega(A/\Delta^2)$ and $\Omega(T)$.
>
> Hence, the proposed regret upper bound exhibits a tight dependence on the gap $\Delta$ and $T$ up to logarithmic factors for both regret and query complexity. However, we are not sure if the dependence on $d$ is tight. Further improvement on either the upper bound or the lower bound is an interesting future direction.
>
> **4. The meaning of $P_t[a_t,b_t]$ on line 146**
>
> On line 146, we are making a comparison to [2], where the notation $P_t$ is introduced, representing the preference matrices in their paper. We will add this definition in the next version to enhance clarity.
>
> **5. Connections to prior RL work which makes use of eluder dimension**
>
> Thanks for pointing this out. We will incorporate a more comprehensive discussion on prior works which use the eluder dimension in the next version. Additionally, we highlight that our worst-case regret bound is independent of the eluder dimension, while most of the existing RL works that use the eluder dimension will have eluder dimension in their regret bound.
>
> ---
>
> [1] Foster, Dylan, et al. "Instance-Dependent Complexity of Contextual Bandits and Reinforcement Learning: A Disagreement-Based Perspective." *Conference on Learning Theory*. PMLR, 2021.
>
> [2] Saha, Aadirupa, and Akshay Krishnamurthy. "Efficient and optimal algorithms for contextual dueling bandits under realizability." *International Conference on Algorithmic Learning Theory*. PMLR, 2022.

---

### Official Review · Reviewer_FECY · 2023-07-07

**Soundness:** 3 good
**Presentation:** 4 excellent
**Contribution:** 3 good
**Rating:** 7
**Confidence:** 3

**Summary:**

This paper develops the provably efficient algorithms AURORA and AURORAE, which are able to achieve the optimal regret bound under contextual dueling bandit setting, and imitation learning respectively, at the same time minimizing query complexity. The key idea behind is that the algorithm only makes a query when the algorithm is very uncertain about the optimal action ($Z_t 1_{|A_t| > 1}$). The algorithm decides the sampling distribution of action pairs to make a query by considering whether the estimated cumulative regret exceeds the carefully designed threshold. If it does not exceed, the algorithm does exploration and sample action pairs from the uniform distribution. If it exceeds, the algorithm uses a technique similar to inverse gap weighting to achieve better balance between exploration and exploitation. For imitation setting with horizon H, the algorithm treats MDP as a concatenation of H contextual bandits and runs AURORAE, which is a stack of multiple AURORA instances.

**Strengths:**

This work is original and well-motivated. It is crucial to design an online learning algorithm that achieves optimal regret while using minimal query complexity. Although I did not get a chance to read the complete proofs in the supplementary material carefully, given the discussion of intuition, all technical results seem reasonable to me.

This paper is well presented and is a pleasure to read. An example for illustration follows every definition. All materials are well organized in a logical manner.


**Weaknesses:**

I have several concerns regarding the proposed algorithms. First, P5 I 5, the computational complexity for the candidate arm set might be very large, even if F is assumed to be a d-dimensional linear class. The computational complexity might be $O(dT\log(T)|A|)$. Also, in reality, F might be very complex, which might even worsen the computational complexity. Can we use a simple function class F for approximation while still achieving a similar regret bound?


**Questions:**

Please see the review in weaknesses.

**Limitations:**

The authors address their limitations of not having any experiments on real data or simulations. I believe the work will be much more convincing if the theoretical bounds are supported by experiment results.

---

> ### Author Rebuttal · Authors · 2023-08-09
>
> Thank you for your valuable feedback! We would like to address your concerns as follows.
>
> **The computational complexity for the candidate arm set**
>
> As mentioned by the reviewer, if $\mathcal{F}$ is a $d$-dimensional linear class, the computational complexity will be $\tilde{O}(d T A)$. We believe that this complexity is true and acceptable, considering that even the well-established linUCB algorithm [1] also exhibits a computational complexity of at least $\tilde{O}(dTA)$ (noting that for linUCB, the construction of the upper confidence bound at each round $t$ takes at least $\tilde{O}(dA)$ time).
>
> With regard to the question "Can we use a simple function class F for approximation while still achieving a similar regret bound?", we apologize we do not fully understand it. Could you clarify what you meant by using a function class for approximation?
>
> ---
>
> [1] Li, Lihong, et al. "A contextual-bandit approach to personalized news article recommendation." Proceedings of the 19th international conference on World wide web. 2010.

---

> > ### Comment · Reviewer_FECY · 2023-08-17
> > **Response**
> >
> > Thanks for the response. Could you discuss more about how specific choices of the function class $\mathcal{F}$ affect the computational complexity and the regret bound?

---

> > > ### Author Response · Authors · 2023-08-19
> > >
> > > Thanks for your response. Below we discuss more about the effect of the choice of function class $\mathcal{F}$ on the computational complexity and the regret bound.
> > >
> > > **How function class affects the computational complexity**
> > >
> > > We observe that the computational complexity of the proposed algorithm mainly depends on the computation of the candidate arm set (Algorithm 1, Line 5) and the width (Line 8).
> > >
> > > When $\mathcal{F}$ is a $d$-dimensional linear class, the computational complexity is $\tilde{O}(d T A)$ since the version space exhibits an ellipsoid structure where both the candidate arm and the width can be solved in $\tilde{O}(d A)$ time. When $\mathcal{F}$ is tabular, it can be considered as a special case of linear class with one-hot encoding. In this case, we have $d=S\times A$, resulting in a computational complexity of $\tilde{O}(TSA^2)$.
> > >
> > > For a more general convex function class $\mathcal{F}$, we can design an efficient algorithm based on a weighted regression oracle for $\mathcal{F}$. We first note that previous work [1] has proposed a method to efficiently compute the width. Now we propose the following method to compute the candidate arm set. We first note that an arm $a$ belongs to the candidate arm set at round $t$ if and only if
> > > $$
> > > \min\_{f\in\mathcal{F},\\,\xi\in\mathbb{R}^A}
> > > \\; 1
> > > \quad\text{s.t.}\quad
> > > f(x,a,a')=\xi\_{a'}\\;
> > > ,\quad
> > > \xi\_{a'} > 0
> > > \quad(\forall a'\neq a)
> > > \quad\text{and}\quad
> > > \sum\_{s=1}^{t-1} Z\_s\left(f(x\_s,a\_s,b\_s)-f\_t(x\_s,a\_s,b\_s)\right)^2 \leq \beta
> > > $$
> > > is feasible. Here we introduce the slack variable $\xi$ so that the optimization part for $f$ can be simply reduced to a weighted regression oracle. Next, we convert the above into Lagrangian formulation and obtain
> > > $$
> > > \begin{aligned}
> > > &\min\_{f\in\mathcal{F},\\,\xi\in\mathbb{R}^A}
> > > \max\_{\alpha\in\mathbb{R}\_+^A,\gamma\in\mathbb{R}\_+^A,\lambda\in\mathbb{R}\_+}
> > > \\; 1 + \sum\_{a'\neq a}\alpha\_{a'}\big(f(x,a,a')-\xi\_{a'}\big)^2 - \sum\_{a'\neq a}\gamma\_{a'} \xi\_{a'}
> > > +\lambda\left(\sum\_{s=1}^{t-1} Z\_s\left(f(x\_s,a\_s,b\_s)-f\_t(x\_s,a\_s,b\_s)\right)^2 - \beta\right)
> > > \\\\
> > > =&
> > > \max\_{\alpha\in\mathbb{R}\_+^A,\gamma\in\mathbb{R}\_+^A,\lambda\in\mathbb{R}\_+}
> > > \min\_{f\in\mathcal{F},\\,\xi\in\mathbb{R}^A}
> > > \\; 1 + \sum\_{a'\neq a}\alpha\_{a'}\big(f(x,a,a')-\xi\_{a'}\big)^2 - \sum\_{a'\neq a}\gamma\_{a'} \xi\_{a'}
> > > +\lambda\left(\sum\_{s=1}^{t-1} Z\_s\left(f(x\_s,a\_s,b\_s)-f\_t(x\_s,a\_s,b\_s)\right)^2 - \beta\right)
> > > \end{aligned}
> > > $$
> > > Here we note that we can swap the min and max since the objective is convex in the joint space of $f$ and $\xi$. Then, the inner minimization problem can be solved by updating $f$ via the regression oracle and updating $\xi$ via gradient descent; for the outer maximization problem, we can do projected gradient ascent.
> > >
> > > **How function class affects the regret bound**
> > >
> > > The function class affects the regret bound in $\beta$ and the eluder dimension. We elaborate on them separately:
> > >
> > > - $\beta$: for commonly used loss function (see Example 2 and 3), $\beta$ depends on the complexity of the class  $\mathcal{F}$, as is standard in oracle-based bounds. However, this dependence typically only scales logarithmically with the size (or effective size) of $\mathcal{F}$ and is mild in many scenarios. Eg. for a finite class $\mathcal{F}$, $\beta$ depends on $\log |\mathcal{F}|$. Furthermore, when $\mathcal{F}$ is infinite, we can replace $|\mathcal{F}|$ by the covering number of $|\mathcal{F}|$ following the standard techniques. To give some concrete examples of infinite classes, for $d$-dimensional linear function class, $\beta$ will have a dependence of $O(d)$ (effective complexity of $\mathcal{F}$), and for the tabular class, $\beta$ will have a  dependence of $O(SA^2)$.
> > >
> > > - Eluder dimension: it is a standard complexity measure of function classes. For linear function class $\mathcal{F}$, it is typically bounded by $d$; for tabular function class, it is bounded by $SA^2$. More examples can be found in existing works on eluder dimension such as [2].
> > >
> > > We will add these explanations to the revised version.
> > >
> > > ---
> > >
> > > [1] Foster, Dylan, et al. "Practical contextual bandits with regression oracles." International Conference on Machine Learning. PMLR, 2018.
> > >
> > > [2] Russo, Daniel, and Benjamin Van Roy. "Eluder dimension and the sample complexity of optimistic exploration." Advances in Neural Information Processing Systems 26 (2013).

---

### Decision · Program_Chairs · 2023-09-21

**Decision:**

Accept (poster)

**Comment:**

All reviewers agree that the problem studied in this paper is well motivated, that the paper is well written and that the technical results are sufficiently novel.


There were slight concerns about the computational complexity of the proposed algorithms and the existence of an efficient regression oracle with meaningful regret guarantees. The authors address these issues in their rebuttal to the reviewers and are encouraged to add the respective discussions in the camera ready version of the paper.
Some reviewers also raised the point of sub-optimality of the presented regret bounds and how the bounds compare with prior work. The authors address these issues to the satisfaction of the reviewers as well.


Overall this is a good paper with sufficiently novel contributions which merits acceptance to the program.